# Emergency Medical Interventions in Areas with High Air Pollution: A Case Study from Małopolska Voivodeship, Poland

Ewa Szewczyk [1], Michał Lupa [2,*], Mateusz Zaręba [1], Elżbieta Węglińska [1], Tomasz Danek [1] and Amit Kumar Mishra [1,3]

1 Faculty of Geology, Geophysics and Environmental Protection, AGH University of Krakow, 30-059 Kraków, Poland; eszewczyk@agh.edu.pl (E.S.); zareba@agh.edu.pl (M.Z.); weglinska@agh.edu.pl (E.W.); tdanek@agh.edu.pl (T.D.); amit@agh.edu.pl (A.K.M.)
2 Faculty of Space Technologies, AGH University of Krakow, 30-059 Kraków, Poland
3 Division of Computer Engineering and Computer Science, University West, 461 32 Trollhättan, Sweden
* Correspondence: mlupa@agh.edu.pl

**Abstract**

Air pollution poses a significant threat to public health, particularly in urban and industrialized regions. This study investigates the relationship between air quality and the frequency of Emergency Medical Service (EMS) calls in the Małopolska Voivodeship of Poland between 2020 and 2023. Data from over 190 air quality sensors ($PM_{10}$) were spatially aggregated using both hexagonal grids and administrative boundaries, while EMS call records were filtered to focus on cardiovascular and respiratory incidents. During 2020–2023, a total of 305,142 EMS calls were analyzed, and months with $PM_{10}$ exceedances showed an average of 1.50 respiratory calls per 1000 residents compared to 1.19 in months without exceedances. Statistical analyses, including Kolmogorov-Smirnov tests and Pearson correlation, were applied to explore temporal and spatial associations. Results indicate a statistically significant increase in EMS calls during periods of elevated air pollution, with the strongest correlation observed for respiratory-related incidents. Comparative analyses between high- and low-pollution municipalities supported the observed relationships. Further analysis indicated that the COVID-19 pandemic may have partially confounded these associations, particularly for respiratory cases, though significant patterns remained even after accounting for pandemic peaks. While limitations related to data gaps and seasonal biases exist, the findings suggest that real-time air pollution data could inform better EMS resource allocation. This research highlights the potential of integrating environmental data into public health strategies to improve emergency response and reduce health risks in polluted regions.

**Keywords:** air pollution; emergency medical services; respiratory diseases; cardiovascular health; spatial analysis; public health; $PM_{10}$; COVID-19

## 1. Introduction

Air pollution is a growing global concern with serious implications for public health, particularly in densely populated or industrialized areas [1–3]. Numerous studies have highlighted the impact of air pollutants, both fine ($PM_{2.5}$) and coarse ($PM_{10}$) particular matter on the respiratory and cardiovascular systems [4–6]. For instance, Michikawa et al. [4] demonstrated a significant association between short-term exposure to PM and an increase in ambulance dispatches in Japan. Similarly, Shi et al. [5] showed that multiple pollutants

interact synergistically to raise the risk of medical emergencies. Carroll et al. [6] went further, linking $PM_{2.5}$ exposure to increased ambulance usage for mental health conditions.

Studies assessing the impact of industrial and mining activities on public health have been conducted, for example, in Australia, where Hendryx et al. [7] demonstrated that coal mining was responsible for 42.1% of national $PM_{10}$ emissions, negatively affecting the health of residents in mining communities, where exposure to $PM_{10}$ was significantly higher than in non-mining areas. Also, studies done in Poland showed that coal-based energy sources are significantly associated with increased PM concentrations, while renewable energy sources contribute to reductions in air pollution, with regional differences linked to industrialization levels and green area coverage [8].

At the Voivodeship level, due to its specific geographical conditions, Małopolska is partially exposed to solid fuel heating pollution—especially during winter extreme smog episodes (ESE) [9,10]. In particular, Kraków (the regional capital and largest urban agglomeration) is situated in a topographical basin that promotes pollutant accumulation, especially during winter months when temperature inversions inhibit air circulation and trap harmful substances close to the ground. Moreover, research has shown that concentrations of particulate matter in Krakow are generally decreasing over years but there are ESEs with alarmingly high pollution levels [10], primarily due to emissions from both local heating systems located outside of the city and transboundary pollution from Silesia, where industrial activity is prevalent [11,12].

Such levels of air pollution have a significant impact on cardiovascular and respiratory health, as evidenced by Naddafi et al. [13], who estimated 2194 excess annual deaths in Tehran attributable to $PM_{10}$, with additional mortality linked to $SO_2$, $NO_2$, and $O_3$. Similarly, Zheng et al. [14] demonstrated that interquartile increases in $PM_{2.5}$ and $PM_{10}$ concentrations were associated with 2.09% and 2.33% increases in hospitalizations for cardiac arrhythmia, respectively, across 26 Chinese cities. Moreover, recent studies have also explored spatio-temporal modeling of air pollutants, for example using VAR-tree machine learning frameworks to predict ozone concentration patterns across large regions of China, effectively capturing complex seasonal and spatial variations [15,16], which also have significant impacts on cardiovascular health.

Studies from Southeast Asia reinforce these findings: Mueller et al. [17] found that same-day $PM_{10}$ exposure from biomass burning in northern Thailand increased outpatient visits for chronic lower respiratory and cerebrovascular diseases. Long-term impacts have also been widely documented. Jacobs et al. [18] highlighted associations between $PM_{10}$ exposure and congenital anomalies in China, while Renzi et al. [19] reported that each 1 $\mu g/m^3$ increase in $PM_{10}$ in the Latium region of Italy corresponded to 0.8% higher nonaccidental, 0.9% cardiovascular, and 1.4% respiratory mortality rates. Consistently, Urbanowicz et al. [20] demonstrated that chronic $PM_{10}$ exposure accelerates the progression of coronary artery disease, underscoring the need for stricter air quality standards and ongoing health surveillance. Air pollution has also been strongly associated with cardiovascular complications. Research by Bhatnagar [21] linked PM exposure to thrombosis, endothelial dysfunction, and systemic inflammation.

Despite the known health risks, few studies have directly linked chronic air pollution to emergency medical service (EMS) demand. We analyse EMS calls as an operational health outcome and characterise exposure at the municipality level by integrating reference stations and dense low-cost sensor networks, using the number of days with $PM_{10}$ exceedances, which captures episodic winter smog and local heterogeneity better than annual averages. This contrasts with Alyami et al. [22], who estimated area-wide exposure from annual mean $NO_2/PM_{2.5}$ assigned from the nearest station together with proximity to major roadways (<1 km; 1–3 km; >3 km), and with Lavigne et al. [23], who

related daily pollutant concentrations to asthma ED visits in a single urban region. Our design therefore contributes: (i) an EMS-linked outcome rather than clinical/ED endpoints; (ii) municipality-level, multi-network exposure based on $PM_{10}$ exceedance days (rather than annual means/proximity); and (iii) a multi-year, pandemic-aware analysis of seasonality across Małopolska (2020–2023), yielding operationally actionable evidence for EMS resource allocation focused on respiratory and cardiovascular calls.

Moreover, the present study addresses this gap by analyzing four years (2020–2023) of data on Emergency Medical Services (EMS) calls and air pollution levels across the entire Małopolska region. Using high-resolution spatial and temporal data on $PM_{10}$ pollution and EMS interventions, we assess whether real-time environmental monitoring can support health service planning and resource allocation. In particular, we focus on ambulance calls related to respiratory and cardiovascular conditions, which are likely to be influenced by air quality.

Our findings aim to provide actionable insights for regional health authorities and policy-makers, offering a scalable framework for integrating air quality monitoring into public health decision-making.

## 2. Materials and Methods

### 2.1. Study Area Overview

The Małopolska (Lesser Poland) Voivodeship is located in southern Poland and covers an area of approximately 15,200 square kilometers (Figure 1). Despite its relatively small size, it is one of the most densely populated and economically significant regions in the country, with around 3.4 million inhabitants. The regional capital, Kraków, is both the largest urban center and a major cultural and economic hub. Geographically, the region is characterized by varied topography including mountain ranges and basins. The geographical structure, particularly the topography and the location of the city within the Vistula Valley, facilitates the influx of pollutants from neighboring cities. The valley's shape acts as a natural corridor, channeling air masses along the Vistula River and contributing to the accumulation and transport of atmospheric pollutants into the urban area [24]. What is more, Kraków is situated in a basin that contributes to frequent temperature inversion events. These inversions, particularly common in winter, hinder air circulation and facilitate the accumulation of air pollutants near the surface, exacerbating local air quality issues [25,26]. Ongoing regional policies target reduction of pollution through measures such as the phased elimination of low-efficiency coal boilers and support for cleaner energy alternatives, including geothermal heating, which shows promise in Małopolska due to substantial geothermal resources. However, challenges remain in accelerating fuel-switching among residents and enhancing enforcement of air quality regulations.

The population distribution in Małopolska spans from densely populated urban areas such as Kraków to less populated rural zones, including popular tourist destinations in the Tatra Mountains and surrounding counties.

For clarity, the spatial units analyzed in this study correspond to three levels of Poland's administrative division, which are commonly used for public health and environmental statistics:

- Municipalities are the smallest administrative units, with populations ranging from approximately 2000 in rural areas to over 750,000 in the city of Kraków, and areas typically between 50 and 150 $km^2$.
- Counties aggregate several municipalities, with populations between 50,000 and 400,000 and areas from 500 to 1500 $km^2$.
- Districts apply only to Kraków, subdividing the city into 18 units with populations from 15,000 to 130,000, providing high urban spatial granularity.

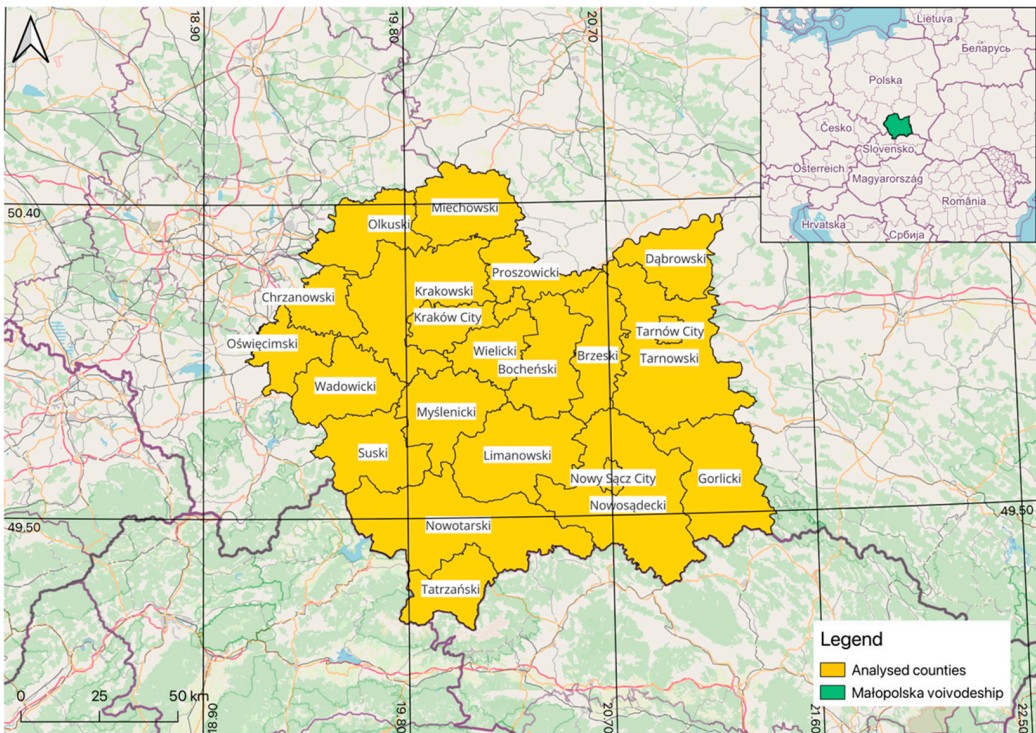

**Figure 1.** Study area: selected counties in Małopolska, with a locator inset marking the voivodeship in green.

This hierarchical structure enabled multi-scale analyses where smaller units (municipalities and districts) captured localized pollution effects, while counties supported the integration of demographic and environmental data for broader population-level assessments.

To ensure consistency across all years of analysis (2020–2023), the population of 2023 was used as the reference year for normalization. According to official demographic statistics from Statistics Poland [27], the total population of Małopolska changed by less than 1% between 2020 and 2023 (3.41 million vs. 3.39 million residents). This minimal variation does not materially affect the calculation of EMS call rates per 1000 residents, making 2023 a valid baseline year for population normalization.

Moreover, Poland's healthcare system, funded through the National Health Fund, provides universal coverage with EMS integrated as a critical component. EMS teams operate under national standards, deploying both basic and specialized units coordinated by regional dispatch centers. In 2022, Małopolska reported EMS call rates above the national average, driven by urban density and seasonal tourist influxes, which place additional strain on emergency response capacity.

### 2.2. Air Pollution Data Sources and Characteristics

Air pollution is commonly measured using particulate matter indicators such as $PM_{2.5}$ and $PM_{10}$, which denote airborne particles with diameters smaller than 2.5 μm and 10 μm, respectively. $PM_{10}$ is particularly useful in public health studies as it includes both finer and coarser particles, allowing a single variable to represent a broad pollution profile [28]. In this study, $PM_{10}$ was selected as the core metric due to its wide availability and standardization across sensors.

Air quality data were obtained from three main sources: (1) the Chief Inspectorate of Environmental Protection (GIOŚ), a public body maintaining 268 nationwide reference stations; (2) Airly (www.airly.com), a private company offering real-time data through

commercially installed low-cost sensors (LCS); and (3) LookO2, a provider of historical air quality data (also LCS). From these, 192 sensors across the Małopolska Voivodeship were used: 28 from GIOŚ, 87 from Airly (concentrated in Kraków and its surroundings), and 77 from LookO2. Due to varying sensor models and data quality, initial validation included a comparative analysis of co-located sensors. While LookO2 sensors exhibited greater signal noise and a higher number of outliers, general temporal trends remained consistent across providers. Therefore, all data were retained after removing outliers exceeding three standard deviations from the yearly mean. It is important to mention that low-cost sensors (LCS) are typically associated with higher uncertainties; however, they are inexpensive and provide the possibility of achieving high spatial coverage. It is also important to note that while LCS sensors are generally associated with higher uncertainties, they are cost-effective and allow for high spatial coverage, something that is difficult to achieve with reference-grade gravimetric manual measurements typically used for reporting [9,10].

The final dataset comprised 3,543,778 hourly records from 192 sensors over four years (01/01/2020–31/12/2023). Of these, 19 GIOŚ sensors provided complete records for the full period. Missing values were not interpolated due to concerns about spatial specificity and pollution localization, especially for gaps exceeding 24 h. Spatial sensor distribution was non-uniform, with a notable concentration around Kraków and gaps near the Limanowa, Myślenice, and Bochnia counties. Figure 2 illustrates the sensor locations and source attribution.

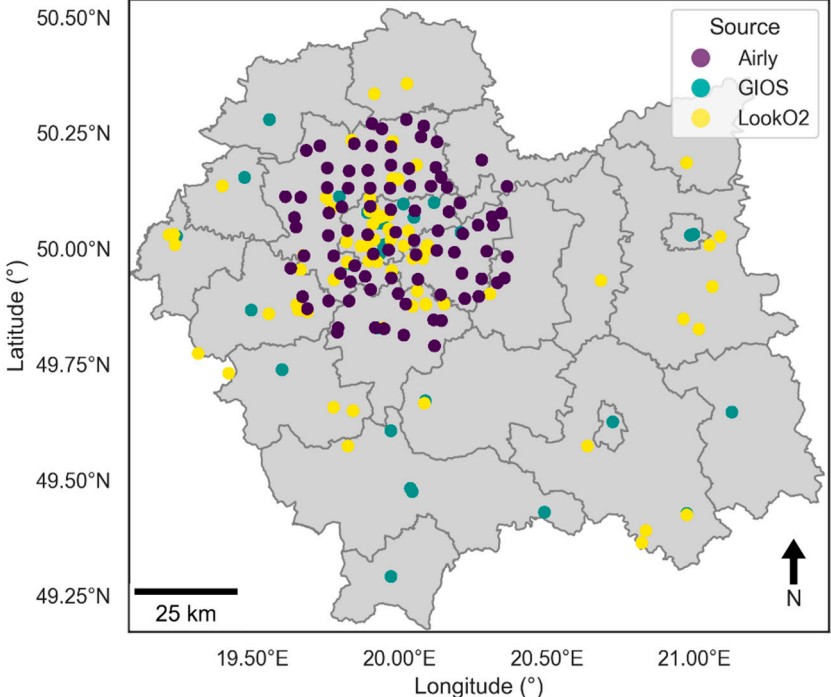

**Figure 2.** Distribution of air quality sensors in Małopolska Voivodeship, distinguished by their source of origin.

## 2.3. EMS Data Processing

Emergency Medical Services (EMS) call data (2020–2023) were obtained from the Małopolska Voivodeship Governor's Office in a tabular, anonymized format, where each record represented a single ambulance dispatch. Each record included the geographic coordinates of the incident location and a primary reason for the call.

Using these coordinates, each EMS call was geocoded and assigned to the corresponding municipality or Kraków district, which enabled spatial aggregation at the local

administrative level. Based on the official call classification codes, each dispatch was categorized into two groups relevant to air pollution health impacts:

- cardiovascular (e.g., cardiac arrest, acute coronary syndrome, hypertension),
- respiratory (e.g., dyspnea, chest pain).

This processing allowed for both spatial aggregation and temporal alignment of EMS calls with $PM_{10}$ exceedance days for subsequent statistical analyses.

### 2.4. Data Processing and Spatial Aggregation

Given the spatio-temporal nature of the dataset, preprocessing involved both spatial and temporal aggregation. The data were treated as two-dimensional: space and time. To enable statistical comparisons across regions and time intervals, spatial aggregation was required. Point-based sensor data were insufficient for area-wide analysis due to uneven sensor distribution.

Outlier detection and removal were conducted prior to aggregation. Hourly values were aggregated into daily averages and then linked to the respective spatial unit. This ensured that short-term fluctuations were smoothed, and the temporal resolution matched EMS response patterns. The aggregation enabled further statistical modeling of relationships between air pollution levels and emergency service demand.

In addition to regular spatial grids, irregular spatial aggregation based on existing administrative units was employed to analyze air quality data. This approach leveraged the natural delineation of Poland's territorial divisions, specifically counties and municipalities. Although these units vary in shape and size, the differences are not substantial enough to significantly impact generalizability. Administrative boundaries often align with geographical features such as rivers and share similar regulatory frameworks (e.g., regional coal-heating bans) and urban-rural typologies, which supports their use in spatial public health studies.

Air quality sensors were present in every county, though not in every municipality. Aggregation at the municipal level provides higher spatial resolution and minimizes generalization error in pollution estimates. However, county-level aggregation is more practical in some cases due to the availability of auxiliary variables such as population density or green space coverage, which are more commonly published at that administrative level.

The threshold-based approach was selected because it is well suited for public health studies that focus on acute outcomes. By classifying each day as either an exceedance or a non-exceedance day according to the WHO 24-h $PM_{10}$ guideline (50 μg/m$^3$) [29], this method captures the persistence of harmful pollution events rather than isolated short-term fluctuations. We did not perform a formal intercalibration of the data sources (GIOŚ, LookO2, Airly). Instead, we minimized cross-network scale differences by computing exceedances from daily means after sensor-level outlier filtering and by aggregating at the municipality level using an any-sensor > 50 μg/m$^3$ rule. This approach is robust to outliers and to variations between low-cost and reference sensors, as individual extreme measurements have limited impact on the binary classification of exceedance days. Moreover, it aligns directly with regulatory and public-health thresholds, facilitating interpretation and supporting its applicability for EMS response planning.

### 2.5. Pollution Assessment Methods

To evaluate the impact of air pollution on public health, a threshold-based approach was implemented using the World Health Organization (WHO) daily air quality guideline for $PM_{10}$, which recommends a maximum 24-h average concentration of 50 μg/m$^3$ [29]. This method focused on counting the number of days in each month when this threshold was exceeded.

First, the data were temporally aggregated by full calendar days for each sensor. If any sensor within a given spatial unit recorded a $PM_{10}$ value above the 50 µg/m$^3$ limit during a 24-h period, that day was classified as an exceedance day for the entire area. This binary classification was then repeated across all sensors and all days of the study period.

Spatial aggregation was performed at the municipal level. Importantly, due to Kraków's unique characteristics including its large population, complex land use (e.g., industrial, historical, and commercial zones), and high pollution levels, it was subdivided into its constituent districts. These districts were treated as equivalent to standard municipalities for the purposes of analysis, as they better reflect population density and local variability in pollution exposure.

The outcome of this method is a monthly count of exceedance days per municipality, ranging from 0 to 31. Compared to average-based methods, this threshold-based approach is more resistant to outliers and better captures the persistence of harmful pollution conditions. An example for January 2023 is presented in Figure 3.

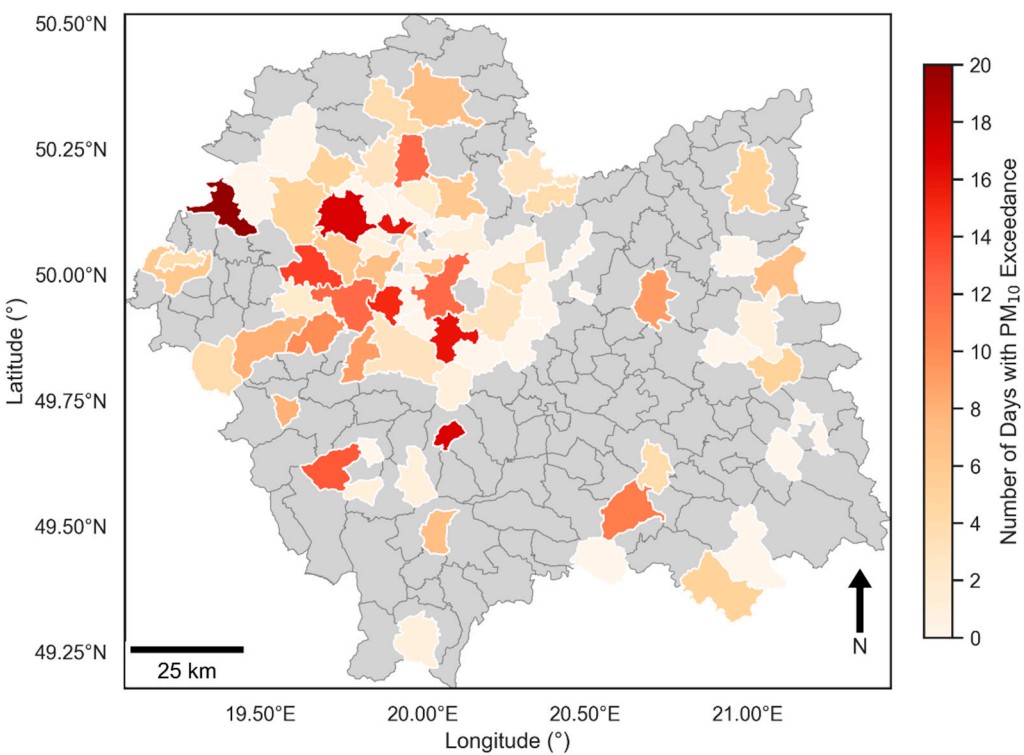

**Figure 3.** Example of the number of days in a month exceeding air quality standards, aggregated by municipality. The chart presents data from January 2023. Areas marked in grey indicate missing data.

It should be noted that in the spatial division at the municipal level, the city of Kraków was separated from other administrative units. Kraków, along with Nowy Sącz and Tarnów, is a city with county rights, whereas in this study, a lower level of spatial aggregation (municipalities) was chosen. The capital of the Lesser Poland Voivodeship covers an area of approximately 327 km$^2$, which is larger than most municipalities. It is also important to note that Kraków has a very large population and various types of land use (industrial, historical, commercial), which could potentially distort the study results.

## 2.6. Criteria and Classification Results for Municipalities

To identify municipalities facing the most severe air pollution issues, a classification was performed based on the number of days in a year when daily $PM_{10}$ concentrations exceeded regulatory limits. In Poland, the legal threshold allows for a maximum of

35 exceedance days per year; thus, any municipality recording 36 or more exceedance days was classified as having a smog problem.

Among the 76 municipalities analyzed, 67 exceeded the annual threshold in at least one year during the study period (2020–2023). Nine municipalities exceeded the limit in all four years, 20 in at least three years, and 21 in two years. These figures are likely underestimated due to data availability issues. In particular, for 2023, Airly sensors only provided data until May 30, reducing the spatial coverage. However, sufficient coverage was retained through data from GIOŚ and LookO2 sensors, ensuring analysis consistency in key locations. Table 1 presents the 15 municipalities with the highest number of exceedance days over the study period.

**Table 1.** Municipalities with the highest number of days exceeding $PM_{10}$ limits per year.

| Municipality | 2020 | 2021 | 2022 | 2023 | Sum |
|---|---|---|---|---|---|
| Mszana Dolna | 128 | 160 | 112 | 43 | 443 |
| Dobczyce | 119 | 111 | 118 | 40 | 388 |
| Skawina | 105 | 101 | 102 | 46 | 355 |
| Oświęcim | 96 | 117 | 93 | 30 | 336 |
| Kalwaria Zebrzydowska | 94 | 120 | 90 | 31 | 335 |
| Wieliczka | 109 | 116 | 80 | 28 | 333 |
| Czernichów | 46 | 132 | 104 | 41 | 323 |
| Dębniki (Kraków) | 81 | 121 | 89 | 28 | 319 |
| Mogilany | 113 | 102 | 65 | 37 | 317 |
| Sucha Beskidzka | 77 | 97 | 92 | 48 | 315 |
| Zabierzów | 67 | 90 | 99 | 52 | 309 |
| Oświęcim (gmina miejska) | 125 | 94 | 51 | 22 | 293 |
| Prądnik Biały (Kraków) | 47 | 94 | 106 | 42 | 289 |
| Stare Miasto (Kraków) | 76 | 89 | 78 | 35 | 278 |
| Zwierzyniec (Kraków) | 102 | 118 | 25 | 30 | 276 |

The next step was to identify municipalities with the cleanest air. This required excluding areas with insufficient data to avoid falsely labeling them as low-pollution zones due to lack of measurements. Municipalities were classified as "low pollution" if:

- They did not exceed the annual 35-day limit in any year, regardless of the number of recorded days.
- At least two years of data were available, with a minimum of 180 days of valid measurements each.

Seven municipalities met these criteria (Table 2). Only one—Gorlice (urban municipality) remained below the threshold in all four years. In its worst year (2021), only 8 exceedance days were recorded across the full 365-day dataset. These results reflect the severity and prevalence of smog in Małopolska Voivodeship.

**Table 2.** Municipalities with the lowest number of $PM_{10}$ exceedance days. "n/a" indicates years with fewer than 180 days of data.

| Gmina | 2020 | 2021 | 2022 | 2023 | Sum |
|---|---|---|---|---|---|
| Gorlice (gmina wiejska) | 1 (365) | 8 (365) | 0 (365) | 2 (365) | 11 |
| Gromnik | 5 (276) | 6 (252) | 1 (347) | n/a (76) | 12 |
| Jordanów (gmina miejska) | 21 (347) | 0 (306) | 0 (331) | n/a (72) | 21 |
| Bochnia (gmina wiejska) | n/a (0) | 28 (300) | 9 (365) | n/a (150) | 37 |
| Wielka Wieś | n/a (0) | 29 (300) | 10 (365) | n/a (150) | 39 |
| Zielonki | n/a (0) | 28 (300) | 16 (365) | n/a (150) | 44 |
| Raciechowice | n/a (0) | 29 (300) | 32 (365) | n/a (150) | 61 |

The calculated exceedance statistics served two main purposes:

(1) Comparative analysis between municipalities with the best and worst air quality (top 7 from each category).

(2) Quantitative classification, where each municipality was assigned a score based on the number of years in which the annual threshold was exceeded. This second classification included only municipalities with at least 180 days of measurements in all four years. Both classification types are visualized in Figure 4.

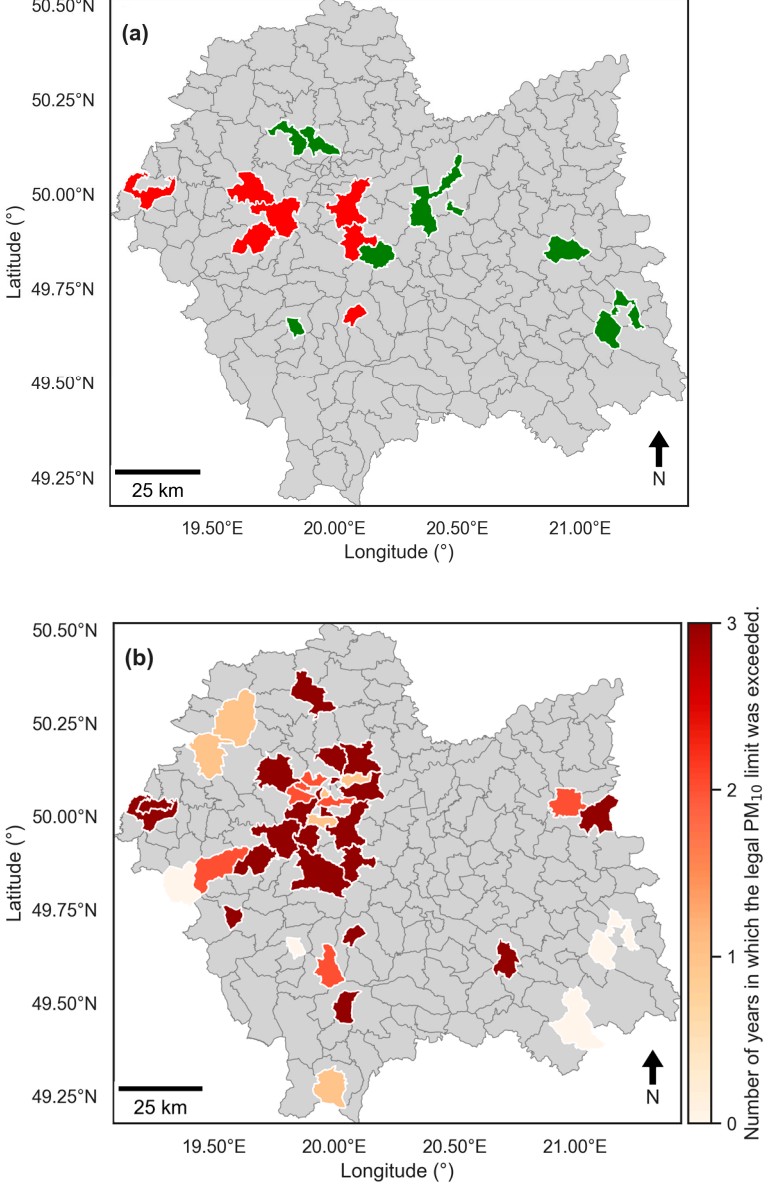

**Figure 4.** Classification results of municipalities in Małopolska Voivodeship based on air quality; (**a**): Seven best (green) and seven worst (red) municipalities during the study period; (**b**): Classification by the number of years in which the legal PM$_{10}$ limit was exceeded. Only municipalities with complete data for all four years were included. Grey areas indicate missing data.

*2.7. Statistical Analysis*

Kolmogorov-Smirnov (KS) and Pearson correlation tests were applied to evaluate the relationship between PM$_{10}$ exceedances and EMS call volumes.

The Kolmogorov-Smirnov test was selected because it allows for a non-parametric comparison of the empirical cumulative distributions of EMS call counts between periods

with and without $PM_{10}$ exceedances, without assuming normality of the data. This makes it particularly suitable for skewed or non-Gaussian distributions common in environmental and health datasets. Because the KS test compares entire empirical distributions rather than mean differences, we interpret results in terms of distributional shifts; means and standard deviations are reported as descriptive context only. Pearson correlation coefficients were used to quantify the linear association between the number of $PM_{10}$ exceedance days and EMS call volumes. This approach is appropriate for identifying the strength and direction of potential relationships across time, supporting the interpretation of temporal patterns observed in the aggregated datasets.

## 3. Results

This section presents the results of the analysis linking Emergency Medical Services (EMS) call volume with air pollution levels and temporal factors. The focus was placed on respiratory and cardiovascular-related incidents occurring within the Małopolska Voivodeship between 2020 and 2023.

*3.1. EMS Call Volume Overview*

A total of 305,142 EMS dispatches related to respiratory and cardiovascular conditions were recorded during the four-year period, corresponding to an average of approximately 208 calls per day. The dataset covered the entire Małopolska Voivodeship and included geolocation, timestamps, and incident categories for each call. To ensure relevance to air pollution effects, the dataset was filtered to include only calls classified as:

- cardiovascular-related: cardiac problems, cardiac arrest, hypertension,
- respiratory-related: shortness of breath, pleuritic chest pain.

The chart below (Figure 5) shows the annual distribution of calls in both categories. A gradual increase in cardiovascular-related calls was observed over the years, while respiratory-related calls peaked in 2021.

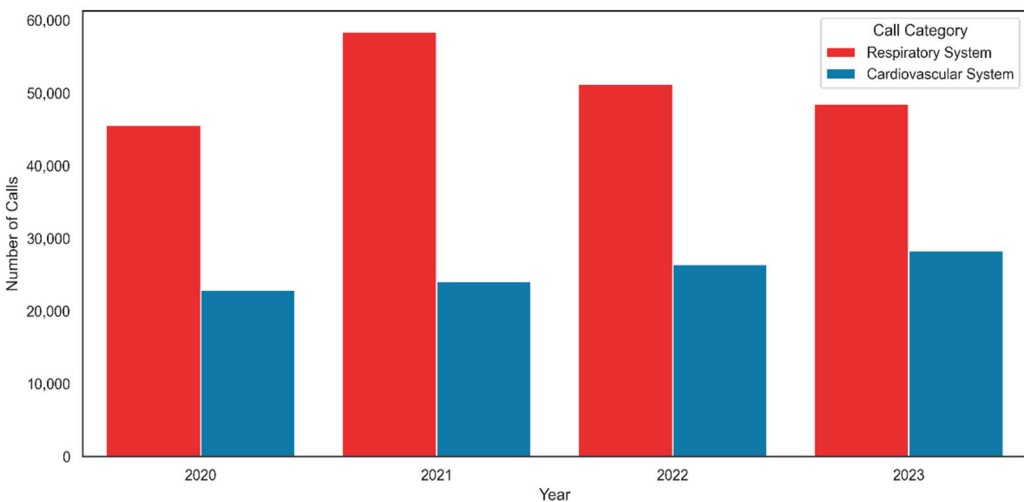

**Figure 5.** Annual number of EMS calls in Małopolska Voivodeship (2020–2023), categorized into cardiovascular and respiratory incidents. Cardiovascular includes: cardiac problems, arrest, and hypertension. Respiratory includes: shortness of breath and chest pain.

*3.2. Data Quality and Aggregation*

The dataset was subjected to preliminary quality assessment. No missing values were found, and exploratory data analysis using histograms and boxplots confirmed the absence of significant outliers. Spatial and temporal aggregation was then performed to align with the air pollution data:

- temporal aggregation: Monthly resolution,
- spatial aggregation: Municipality-level granularity.

Only municipalities for which both EMS and air pollution data were available were retained for final analysis. Therefore, Figure 6 illustrates the outcome of this aggregation, showing the rate of EMS calls per 1000 residents in January 2023.

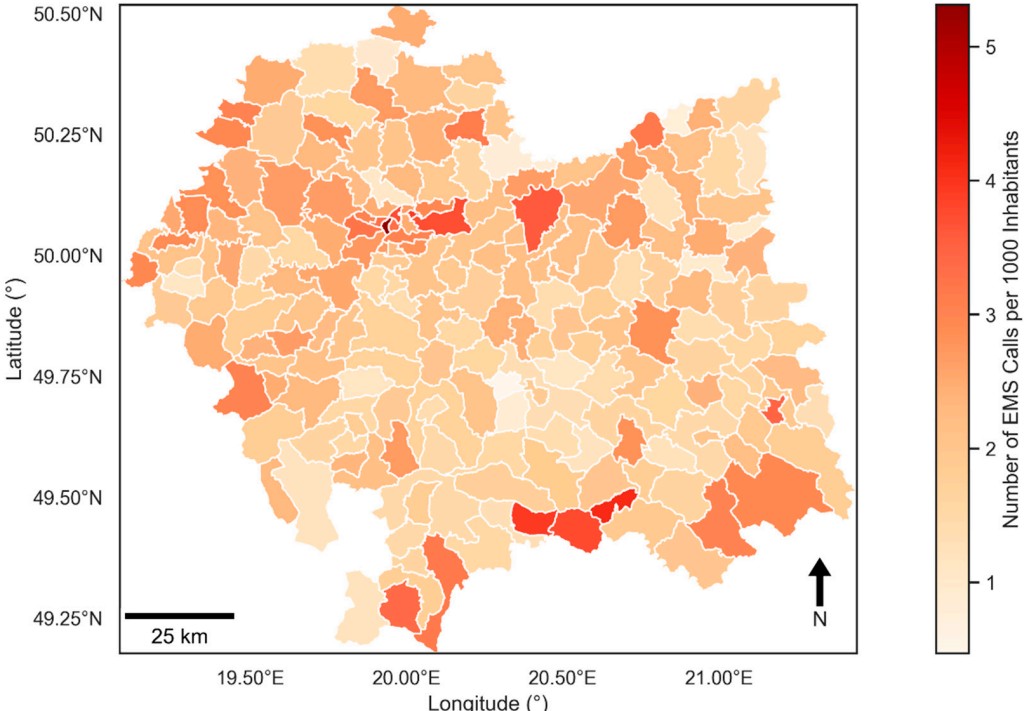

**Figure 6.** Spatial distribution of EMS call rate per 1000 inhabitants in Małopolska Voivodeship (January 2023), including respiratory and cardiovascular incidents.

### 3.3. Impact of Air Pollution on the Number of EMS Calls

A combined dataset integrating information on air pollution levels and the number of EMS (Emergency Medical Services) interventions in individual districts was subjected to statistical analysis, including Kolmogorov-Smirnov tests. The goal was to determine whether there is a general relationship between air quality and the frequency of EMS interventions. In addition, population data for each municipality (as of 1 January 2023) was incorporated, allowing results to be normalized as the number of EMS calls per 1000 inhabitants.

The main research hypothesis tested in this study was: In municipalities with long-term exposure to poor air quality, a stochastically larger distribution of EMS calls can be expected. To verify this hypothesis, both temporal and spatial statistical analyses were conducted. These included general tests across all municipalities with available data, as well as a comparative analysis of municipal groups defined in the previous Section.

#### 3.3.1. General Statistical Analysis

The first test evaluated whether the number of EMS calls differed significantly between months in which $PM_{10}$ air quality limits were never exceeded and those in which exceedances were recorded. The two-sample Kolmogorov-Smirnov test was employed to assess whether the two distributions originated from the same population. Importantly, this test does not assess whether values in one group are greater than in another per se, but whether the empirical cumulative distribution functions (ECDFs) differ significantly in shape or position.

Table 1 and 1408, respectively, representing a roughly even split. The null hypothesis assumed that the distribution for Group 1 (months with no PM$_{10}$ exceedances) was greater than or equal to that of Group 2. This is similar, but not equivalent, to claiming that the values in Group 1 are lower in magnitude. Figure 7 below shows the histograms and ECDFs for both groups across all EMS calls, cardiovascular events, and respiratory events.

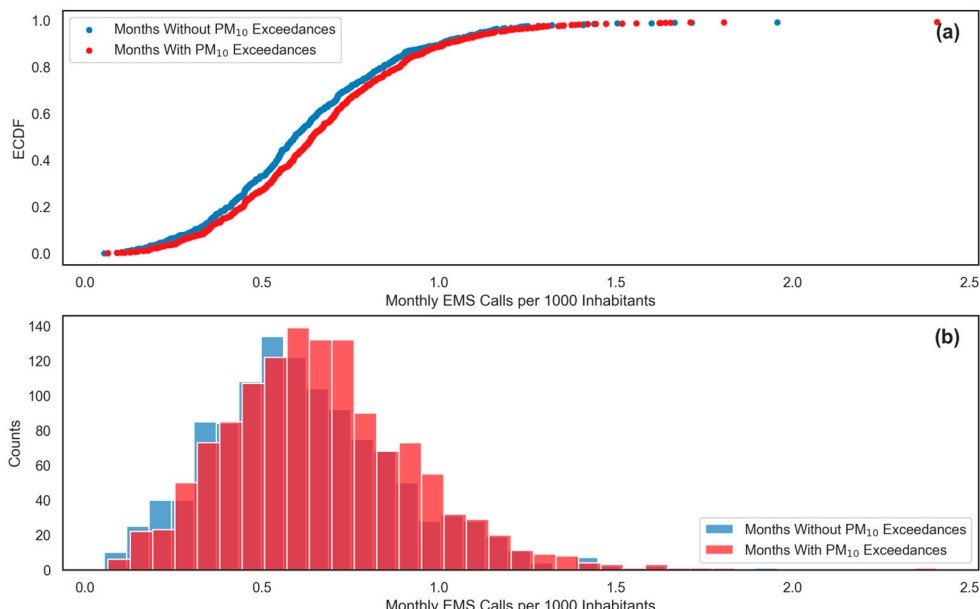

**Figure 7.** Comparison of the number of EMS calls in months with and without PM$_{10}$ exceedances.

The top panel shows the empirical cumulative distribution functions for both data groups, and the bottom panel displays the corresponding histograms.

KS results indicate that EMS calls are stochastically smaller on non-exceedance days, consistent with a right shift of the distribution under exceedance. To confirm this, statistical tests were conducted, with results summarized in Table 3 for all EMS calls, as well as separately for cardiovascular and respiratory events.

**Table 3.** Kolmogorov-Smirnov test results comparing Group 1 (months with no PM$_{10}$ exceedances) to Group 2 (months with exceedances), including basic descriptive statistics.

| Type of Incident | Mean (Group 1) | Std Dev (Group 1) | Mean (Group 2) | Std Dev (Group 2) | KS Test Result | KS Statistic | *p*-Value |
|---|---|---|---|---|---|---|---|
| All calls | 1.8214 | 0.6414 | 2.1696 | 0.7571 | Null hypothesis rejected | 0.2205 | $6.25 \times 10^{-29}$ |
| Cardiovascular | 0.6397 | 0.2677 | 0.6718 | 0.2745 | Null hypothesis rejected | 0.0735 | 0.0007 |
| Respiratory | 1.1872 | 0.4736 | 1.5025 | 0.6216 | Null hypothesis rejected | 0.2490 | $1.19 \times 10^{-36}$ |

Table 2 (months with PM$_{10}$ exceedances) were clearly higher. Although the relationship was not equally strong for all incident types, the results suggest a consistent link between air quality and EMS call volume.

The study also revealed significant temporal variability in EMS call numbers across different periods, with evidence supporting a potential link to changes in air pollution levels. These findings indicate that air quality may indeed affect EMS call rates. To further validate this conclusion, spatial analysis will be carried out in the next section.

### 3.3.2. Comparative Analysis of Municipalities

To further investigate the potential relationship between air quality and EMS call frequency, a comparative analysis was conducted between municipalities classified as

having persistently good and persistently poor air quality. From the dataset, two groups were extracted, consisting of 234 and 279 samples, respectively. The same hypothesis structure was applied: the null hypothesis assumed that EMS call values in the first group (good air quality) originate from a distribution with values greater than or equal to those in the second group (poor air quality). Results of the Kolmogorov-Smirnov test and descriptive statistics for both groups are presented in Table 4.

**Table 4.** Kolmogorov-Smirnov test results comparing Group 1 (municipalities with long-term good air quality) to Group 2 (municipalities with the worst air pollution in Małopolska). Also included are the means and standard deviations for each type of EMS call.

| Type of Incident | Mean (Group 1) | Std Dev (Group 1) | Mean (Group 2) | Std Dev (Group 2) | KS Test Result | KS Statistic | *p*-Value |
|---|---|---|---|---|---|---|---|
| All calls | 1.6609 | 0.6505 | 1.8612 | 0.6513 | Null hypothesis rejected | 0.2197 | $3.53 \times 10^{-6}$ |
| Cardiovascular | 0.5570 | 0.2604 | 0.5830 | 0.2111 | Null hypothesis rejected | 0.1667 | 0.0008 |
| Respiratory | 1.1181 | 0.5524 | 1.2824 | 0.5832 | Null hypothesis rejected | 0.2141 | $6.68 \times 10^{-6}$ |

The results revealed statistically significant differences between the two municipal groups. Specifically, areas with long-term elevated $PM_{10}$ levels showed right-shifted distribution of calls, most notably for respiratory conditions. These findings support the core research hypothesis that increased air pollution is associated with higher EMS demand, particularly for respiratory and cardiovascular emergencies.

For context, the means differed by ~12.1% (1.28 vs. 1.12 per 1000 residents), consistent with the distributional difference detected by KS compared to cleaner municipalities (1.28 vs. 1.12 calls per 1000 residents, $p < 0.001$). The difference for cardiovascular calls was smaller (0.58 vs. 0.56 calls per 1000 residents, $p = 0.0008$). These findings quantitatively confirm that higher pollution areas are associated with greater EMS demand.

Figure 8 illustrates the empirical cumulative distribution functions and histograms for respiratory-related calls in both groups. The graphical comparison further supports the conclusion that the number of EMS interventions was higher in municipalities with poorer air quality. While the results are compelling, caution is warranted in interpreting them. The number of cases analyzed was limited, and the phenomenon is inherently complex and influenced by various confounding factors. Later sections of this thesis delve deeper into additional variables that may mediate or confound the observed relationship.

*3.4. The Role of Temporal and Epidemiological Context*

The World Health Organization (WHO) declared the COVID-19 pandemic on 11 March 2020. Over three years later, on 5 May 2023, the organization stated that COVID-19 was no longer a global public health emergency, although it did not declare the pandemic officially over [30]. As a result, the majority of the analysis period in this study falls within the pandemic timeline, about 80% of the monthly intervals analyzed took place during the pandemic, while only 20% reflect the time before (just over 2 months) and after (more than half a year). Notably, the non-pandemic data covers fewer than 300 days.

COVID-19 had a profound impact on healthcare systems in Poland and worldwide. The virus, which primarily affects the respiratory system, causes symptoms similar to pneumonia and influenza. Since this study includes data on EMS calls related to respiratory distress conditions most influenced by air pollution, it was essential to consider how the pandemic may have disrupted the dataset and influenced the observed relationships. To address this, the research included two key checks:

- Statistical tests were repeated with the exclusion of pandemic "wave" periods.

- Trends in EMS call volume were analyzed across time, both by calendar intervals (months/quarters) and pandemic stages, to evaluate possible seasonal effects and identify pandemic-induced anomalies.

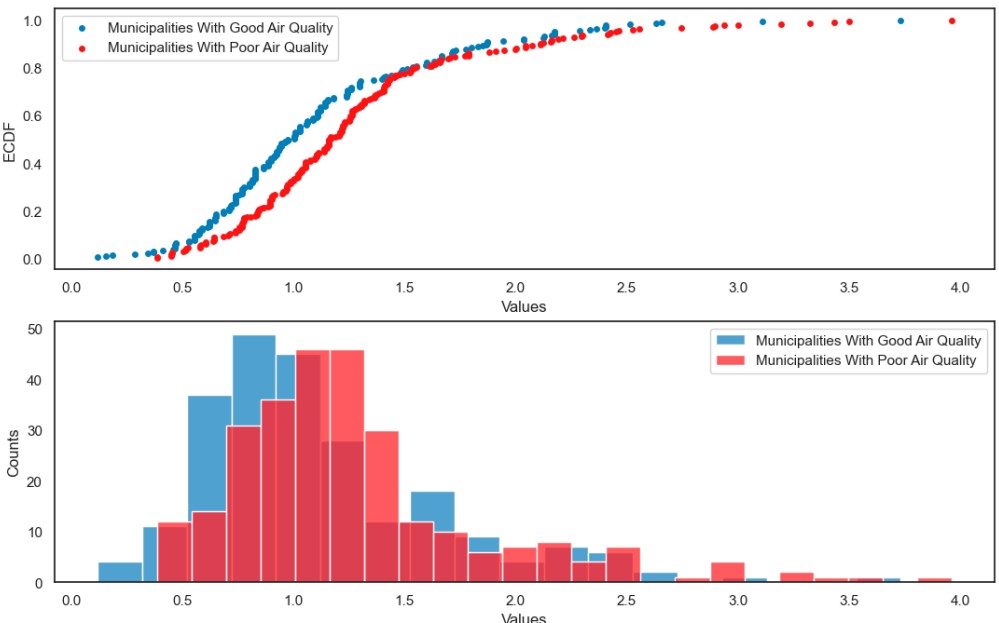

**Figure 8.** Comparison of respiratory-related EMS call volumes between municipalities with good and poor air quality.

3.4.1. Assessing the Influence of COVID-19 on the Results

COVID-19 peak periods were defined based on infection wave maxima identified using the quadratic trend model of daily new cases, following the methodology of Li and Linton [30]. In Poland, these peaks correspond approximately to the periods of March–May 2020, November 2020–January 2021, and December 2021–February 2022 [1].

Initial analyses suggested a correlation between air pollution and EMS calls, particularly those involving respiratory conditions. Since the dataset includes the full duration of the pandemic, there was a need to verify whether this association persisted outside of peak infection periods. Three high-incidence periods were excluded: October–December 2020, March–April 2021 and November 2021–March 2022.

These months coincide with winter, a season associated with poor air quality, which may confound the analysis. The updated statistical results (Table 5) showed that the differences between groups remained statistically significant, albeit with lower certainty evident from the test statistic values and *p*-values compared to earlier results.

**Table 5.** Kolmogorov–Smirnov test results excluding peak COVID-19 periods.

| Call Type | Mean (Group 1) | SD (Group 1) | Mean (Group 2) | SD (Group 2) | KS Test Result | KS Statistic | *p*-Value |
|---|---|---|---|---|---|---|---|
| All | 1.7852 | 0.6097 | 1.9785 | 0.6947 | Reject null hypothesis | 0.1317 | $3.31 \times 10^{-8}$ |
| Cardiovascular | 0.6381 | 0.2670 | 0.6951 | 0.2805 | Reject null hypothesis | 0.0906 | 0.0003 |
| Respiratory | 1.1524 | 0.4350 | 1.2900 | 0.5045 | Reject null hypothesis | 0.1641 | $2.57 \times 10^{-12}$ |

The decrease in correlation strength (especially for respiratory cases) suggests that pandemic-related factors may have distorted earlier findings. Specifically, respiratory-related EMS calls showed a larger variance than cardiovascular ones, which could indicate that pollution is not as directly linked to heart-related emergencies.

### 3.4.2. Aggregated Data Trends

Figure 9 presents trends in monthly EMS calls alongside average monthly $PM_{10}$ exceedance days. Notably, peaks in respiratory-related EMS calls align with pandemic waves. Simultaneously, air pollution follows a seasonal cycle, increasing in winter and decreasing in summer.

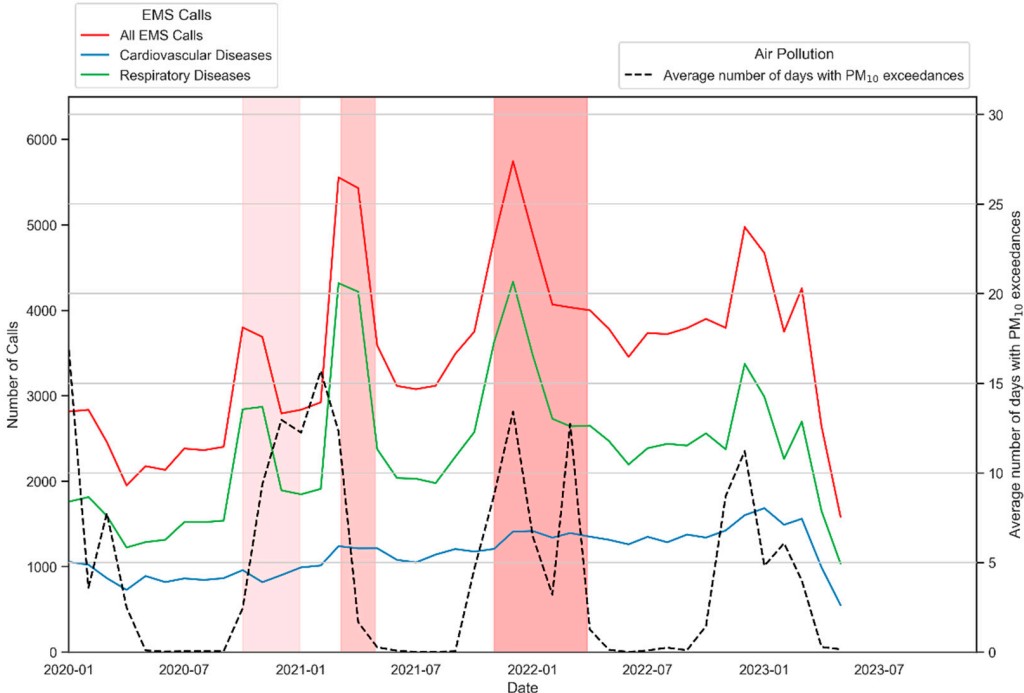

**Figure 9.** The figure presents the monthly average number of $PM_{10}$ exceedance days in relation to EMS calls (both total and by type). Consecutive waves of the COVID-19 pandemic are also indicated, highlighted as red intervals.

Pearson correlation coefficients confirmed a moderate association between air pollution and respiratory-related EMS calls (r = 0.41). However, when pandemic waves were excluded, the correlation dropped to 0.24 (Table 6).

Notably, the cardiovascular-respiratory call correlation increased post-exclusion, while pollution-respiratory correlation declined. This suggests that respiratory issues during the pandemic were more strongly influenced by COVID-19 than by pollution.

**Table 6.** Pearson correlation matrix before (**a**) and after (**b**) removing pandemic wave periods. (**a**) All Data. (**b**) Excluding Pandemic Peaks.

| (a) | | | | |
|---|---|---|---|---|
| | **Pollution** | **Cardio** | **Respiratory** | **All Calls** |
| Pollution | 1.000 | 0.247 | 0.400 | 0.383 |
| Cardiovascular | 0.247 | 1.000 | 0.727 | 0.838 |
| Respiratory | 0.400 | 0.727 | 1.000 | 0.984 |
| All Calls | 0.383 | 0.838 | 0.984 | 1.000 |

**Table 6.** *Cont.*

| | Pollution | Cardio | Respiratory | All Calls |
|---|---|---|---|---|
| | **(b)** | | | |
| Pollution | 1.000 | 0.228 | 0.243 | 0.240 |
| Cardiovascular | 0.228 | 1.000 | 0.957 | 0.981 |
| Respiratory | 0.243 | 0.957 | 1.000 | 0.995 |
| All Calls | 0.240 | 0.981 | 0.995 | 1.000 |

It should be emphasized that removing pandemic months also removed many winter observations, likely periods of poor air quality. This reduction in sample variety may weaken the visible impact of pollution in the data.

### 3.4.3. Seasonality and the Impact of the COVID-19 Pandemic

To assess seasonality, EMS calls were aggregated monthly (Figure 10). However, no consistent seasonal pattern emerged. The clearest increases occurred during pandemic peaks. The first and third quarters of 2023 may reflect post-pandemic trends, but it remains unclear whether the increase in respiratory calls was part of a seasonal pattern or an isolated anomaly.

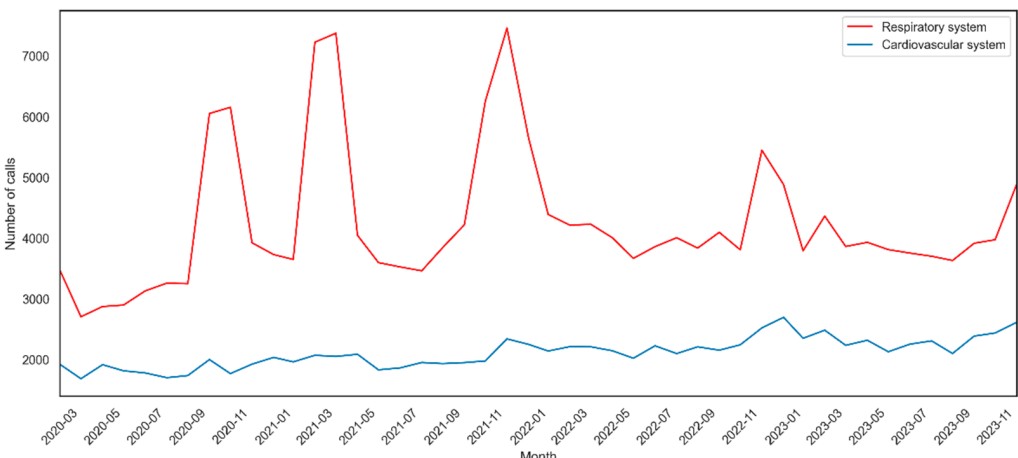

**Figure 10.** Monthly EMS call totals (all categories).

Basic monthly statistics (Table 7) show a clear seasonal pattern. Respiratory EMS calls peak in late autumn/winter—November–December ($\approx$5.0–5.4 k) and remain elevated in January–March ($\approx$3.9–4.8 k), with a trough in June–August ($\approx$3.5–3.6 k). By contrast, cardiovascular calls are comparatively stable across months, varying within a narrower band ($\approx$1.95–2.35 k), with modest winter maxima (January/December) and minima in summer/early autumn (June–September). These monthly results align with Figure 10 and the exceedance-day analysis, indicating that winter pollution episodes coincide with higher respiratory demand, while cardiovascular demand varies less over the year.

**Table 7.** Descriptive statistics by month.

| Month | Respiratory Mean | Respiratory SD | Cardio Mean | Cardio SD |
|---|---|---|---|---|
| January | 4557.0 | 878.2 | 2351.5 | 280.3 |
| February | 3922.2 | 323.1 | 2174.8 | 167.5 |
| March | 4815.2 | 1652.8 | 2169.2 | 239.6 |
| April | 4540.0 | 1996.6 | 2042.8 | 253.8 |

**Table 7.** *Cont.*

| Month | Respiratory Mean | Respiratory SD | Cardio Mean | Cardio SD |
|---|---|---|---|---|
| May | 3711.2 | 562.2 | 2113.5 | 165.8 |
| June | 3490.2 | 406.7 | 1946.2 | 152.3 |
| July | 3564.8 | 322.8 | 2028.0 | 244.5 |
| August | 3604.5 | 322.1 | 2012.2 | 254.8 |
| September | 3637.8 | 278.4 | 1992.2 | 206.2 |
| October | 4568.2 | 995.7 | 2120.0 | 196.5 |
| November | 5045.0 | 1338.3 | 2103.8 | 293.9 |
| December | 5427.2 | 1491.5 | 2347.8 | 305.8 |

3.4.4. Pandemic Phase-Based Analysis

The study period was divided into five distinct pandemic phases to evaluate the influence of COVID-19 on EMS activity. Table 8 summarizes EMS statistics for each phase.

**Table 8.** EMS call statistics by COVID-19 pandemic phase.

| Phase | Dates | Total Calls | Mean/Day | Min/Day | Max/Day | Std. Dev. |
|---|---|---|---|---|---|---|
| Pre-pandemic | 2020-01-01 to 2020-03-03 (62 d) | 13,162 | 208 | 164 | 256 | 22 |
| Pandemic onset | 2020-03-04 to 2021-01-14 (316 d) | 57,917 | 182 | 120 | 402 | 50 |
| Vaccination phase | 2021-01-15 to 2022-05-15 (485 d) | 109,954 | 226 | 130 | 446 | 56 |
| Epidemic threat | 2022-05-16 to 2023-07-01 (411 d) | 86,347 | 209 | 139 | 362 | 31 |
| Post-pandemic recovery | 2023-07-02 to 2023-12-31 (182 d) | 37,762 | 206 | 159 | 313 | 26 |

The pre-pandemic period includes the first two months in the dataset and may be considered a reference point for the subsequent phases. The standard deviation for this group is the lowest among all analyzed periods, although this is likely due to the short time span (62 days).

The initial phase of the pandemic covers a longer timeframe and is characterized by a significant decrease in the average number of daily EMS calls, accompanied by an increase in standard deviation. The range between the minimum and maximum daily call volumes is considerable (282), indicating substantial variability during this phase. This may be explained by the fact that for much of this period, the number of COVID-19 cases in Poland remained relatively low, and people spending more time at home may have contributed to improved overall health. However, the end of this phase coincides with the first wave of COVID-19 in Poland, which occurred in early 2021. During this time, EMS teams were frequently dispatched to patients, resulting in an increased number of calls.

The third phase recorded the highest average number of EMS calls, the greatest range between minimum and maximum daily incidents, and the highest standard deviation. This phase also lasted the longest of all the defined periods. These statistics suggest that it was an exceptionally dynamic time, further supported by the fact that two major pandemic waves occurred during this phase.

The following phase, also covering a long period (411 days), began after the conclusion of the third wave and was characterized by a gradual stabilization of the situation. The average number of daily cases dropped to levels close to those observed in the pre-pandemic phase. Compared to the second and third phases, the standard deviation decreased significantly, further indicating a stabilization of the situation.

The final phase, referred to as post-pandemic recovery, continued the trend of normalization and a reduction in EMS call volumes. Although the average number of calls decreased, the change was not as pronounced as in previous transitions. Notably, the range in daily call numbers, as well as the standard deviation, also decreased, approaching values only slightly higher than those seen in the pre-pandemic period.

Dividing the study period into distinct phases corresponding to the progression of the COVID-19 pandemic made it possible to observe significant differences in both the average number of EMS calls and the variability of those values. Clear fluctuations were visible in the data across average daily counts, standard deviations, and amplitudes. This indicates that the pandemic had a considerable impact on the operations of the emergency medical system. The greatest variability occurred during the most intense phase of the pandemic, while subsequent phases demonstrated a gradual return to stability.

The results of the conducted analysis do not indicate clear seasonality in the number of EMS calls across months or quarters of the year. If such patterns do exist, they were likely disrupted by the effects of the COVID-19 pandemic, which significantly overshadowed natural seasonal fluctuations. Therefore, it is reasonable to conclude that the pandemic had a substantial and complex impact on EMS call volumes by an effect that is difficult to model precisely and should be considered in further analyses, especially when evaluating the influence of environmental factors.

## 4. Discussion

### 4.1. Key Findings

This study investigated the relationship between air quality, particularly the number of days with exceeded $PM_{10}$ pollution limits, and the number of ambulance dispatches (EMS calls) related to respiratory and cardiovascular incidents. The analysis was carried out both spatially and temporally across the Malopolska Voivodeship, using administrative units such as municipalities, Krakow districts, and counties (for demographic analysis).

The results demonstrate a clear association between poor air quality and an increase in EMS calls, particularly for respiratory conditions. While a statistically significant difference was also observed in cardiovascular-related calls during periods and areas of poorer air quality, the strength of this association was notably weaker. This finding is consistent with clinical evidence that greater long-term exposure to traffic-related air pollution is associated with a higher respiratory burden. For example, Alyami et al. [23] reported increased odds of chronic respiratory disease among adults exposed to higher annual $PM_{2.5}$ and $NO_2$, particularly for those living closer to major roadways. Although that study evaluated chronic outcomes rather than EMS demand and relied on annual-mean/proximity metrics, the direction of effect aligns with our observation that respiratory EMS calls are elevated in municipalities with frequent $PM_{10}$ exceedances. By focusing on EMS call categories a priori linked to pollution and using exceedance-based, municipality-level exposure from multiple sensor networks, our approach is tailored to detect acute, operationally relevant effects. Our spatial analysis indicates that municipalities with persistently poor air quality experienced 12.1% more respiratory EMS calls on average compared to cleaner areas (1.28 vs. 1.12 calls per 1000 residents, $p < 0.001$). This magnitude aligns with prior studies reporting increased respiratory risks from particulate matter exposure. Alyami et al. [23] found that adults in high-pollution urban areas had 2.45 times higher odds of chronic respiratory disease and a mean $FEV_1$ reduction of 0.45 L compared to low-exposure populations. Orellano et al. [31] reported that short-term $PM_{10}$ exposure increased the odds of moderate or severe asthma exacerbations by 2–4% (OR 1.024 overall, 1.047 in children). Mrad Nakhlé et al. [32] observed a 1.2% increase in respiratory emergency admissions per 10 $\mu g/m^3$ $PM_{10}$, rising to 1.9% among the elderly. While our study focuses on EMS calls rather than chronic

conditions or single-day hospitalizations, the relative increase aligns with the elevated respiratory burden documented in these prior studies, reinforcing the validity of our spatial findings.

A key difference lies in spatial granularity and exposure assignment. Whereas Alyami et al. [22] estimated exposure area-wide using annual mean $NO_2$/$PM_{2.5}$ from the nearest fixed station combined with proximity to major roadways, and Lavigne et al. [23] related daily concentrations from fixed monitors to ED visits within a single urban region, our study constructs municipality-level exposure from multiple sensor networks and uses $PM_{10}$ exceedance-day metrics. Coarser, station-based assignment and regional aggregation can smooth or misclassify local hot spots, attenuating pollution–health associations; by contrast, our fine-scale approach reveals within-region heterogeneity and sharper contrasts in EMS demand.

Moreover, this study contributes by examining both short-term (days and months with pollution exceedances) and long-term (areas with persistent poor air quality) effects. The identification of a long-term relationship reinforces the argument that chronic exposure to polluted air can have cumulative health effects. This aligns with findings from international studies, such as Wu et al. [33], which showed that long-term exposure increased COVID-19 mortality and complications.

The COVID-19 pandemic played a dual role in this study. On one hand, it increased the number of respiratory-related EMS calls, particularly during the infection waves. On the other hand, lockdown periods temporarily improved air quality, as observed globally [34] potentially reducing the overall pollution-related burden. However, removing the most pandemic-intensive months from our analysis weakened the correlation between air pollution and EMS calls. This suggests a complex interaction between pollution, infection peaks, and seasonal effects. For instance, both air pollution and infectious disease rates rise in winter, confounding direct attributions.

### 4.2. Limitations

Several methodological limitations must be acknowledged. The air quality data did not include topographic or meteorological variables such as wind speed, humidity, atmospheric pressure, or temperature inversion data, all of which can influence pollution dispersion, accumulation, and exposure. This limitation may be particularly relevant in topographically complex regions such as Małopolska, where valleys and basins can trap pollutants under inversion conditions. Incorporating such variables in future analyses could improve the understanding of pollution-EMS relationships.

Health data lacked patient-level attributes (e.g., age, gender, medical history), which could clarify individual risk factors. Additionally, hospital emergency room data (for self-presenting patients) was not incorporated, potentially leading to an underestimation of the overall health impact.

To enhance future research, the deployment of additional data collection framework based on air quality sensors with high spatial coverage combined with satellite measurements is recommended to improve spatial coverage. Ensuring consistency and calibration across sensor models is essential to avoid data heterogeneity. Moreover, integrating meteorological variables and more detailed health and socioeconomic data could refine causal inferences.

Although the multi-year analysis provides robust evidence, the study is not without limitations related to data availability. One limitation of this study is the uneven temporal coverage of Airly sensors in 2023, which provided measurements only until May. This reduced the ability to fully capture exceedance days in the second half of that year. However, because our analysis spans four years (2020–2023) and relies on multi-annual trends, the

incomplete 2023 dataset has only a limited impact on the overall conclusions. Moreover, spatial gaps were partially mitigated by complementary data from GIOŚ and LookO2 sensors, ensuring coverage in key areas. Another source of potential bias arises from the higher noise levels in LookO2 sensors compared to GIOŚ and Airly. Although outliers were removed (values exceeding 3 SD from the annual mean), residual variability may have introduced minor fluctuations in monthly exceedance counts. Importantly, these limitations are unlikely to significantly affect the main findings, as the multi-year analysis, spatial aggregation, and inclusion of multiple sensor networks provide a robust basis for assessing the relationship between air pollution and EMS call volume.

Despite its limitations, this study's results may have practical implications. They could indicate a potential need to consider allocating more advanced ambulance units (e.g., Advanced Life Support, ALS) and specialized respiratory equipment (CPAP, BIPAP, nebulizers, gas analyzers) to municipalities with consistently poor air quality. Such measures could, in theory, enhance emergency response capacity in high-risk areas, but further quantitative assessment of resource needs and potential health benefits is warranted before making definitive recommendations.

In summary, the study confirms a significant link between air pollution and EMS call volumes, especially for respiratory conditions. While the pandemic distorted usual patterns, the underlying association remains evident. By using fine-resolution spatial data and distinguishing short- and long-term effects, this research adds valuable insights to the field. Addressing identified limitations and incorporating broader variables can support better emergency planning and public health policy [35].

### 4.3. Future Work

While the present analysis was based on daily average $PM_{10}$ concentrations in line with WHO guidelines and public health thresholds, evidence from previous research suggests that short-term peaks may have distinct and potentially stronger health impacts. For example, Fu et al. [36] demonstrated that hourly peaks in $PM_{2.5}$ and $NO_2$ concentrations were robustly associated with increased acute myocardial infarction (AMI) hospital admissions in Beijing, using a generalized additive model to capture peak-exposure effects. In contrast, Lin et al. [37] found no significant association between hourly peak $PM_{2.5}$ concentrations and AMI but stressed the importance of considering peak values when evaluating cardiovascular mortality risks. These mixed findings underscore the complexity of short-term exposure assessments and their implications for EMS demand. Future studies could extend this work by incorporating hourly or sub-daily exposure metrics to assess whether such peaks provide additional predictive power for EMS call frequency compared to daily averages.

## 5. Conclusions

This study confirms a significant association between air pollution, particularly $PM_{10}$ exceedance days, and the frequency of Emergency Medical Services (EMS) calls in the Małopolska Voivodeship of Poland between 2020 and 2023. Respiratory-related calls showed the strongest response to periods and areas of elevated air pollution, while cardiovascular-related calls demonstrated a weaker but still significant association.

The multi-year and multi-source analysis, leveraging data from over 190 air quality sensors combined with geolocated EMS records, highlights that municipalities with persistently poor air quality exhibited up to 12% higher respiratory EMS call rates compared to cleaner municipalities. Temporal analysis further showed that pollution peaks and pandemic waves frequently overlapped, complicating causal attribution but confirming a cumulative burden on public health.

These findings underline the practical value of integrating real-time air quality data into emergency planning, enabling more efficient allocation of advanced ambulance units and respiratory support equipment in high-risk areas. Future work should focus on incorporating meteorological, topographic, and patient-level health data to strengthen causal inferences and support proactive, data-driven public health interventions.

**Author Contributions:** Conceptualization, M.L. and E.S.; methodology, E.S. and M.L.; software, E.S.; validation, M.L., M.Z. and E.W.; formal analysis, E.S.; investigation, M.L.; resources, M.Z.; data curation, M.L. and T.D.; writing—original draft preparation, E.S.; writing—review and editing, M.L. and T.D.; visualization, E.S.; supervision, M.L.; project administration, M.L. and A.K.M.; funding acquisition, M.L. All authors have read and agreed to the published version of the manuscript.

**Funding:** This research was funded within the framework of statutory research conducted by the Faculty of Space Technologies and the Faculty of Geology, Geophysics and Environmental Protection at AGH University of Krakow.

**Institutional Review Board Statement:** Not applicable.

**Informed Consent Statement:** Not applicable.

**Data Availability Statement:** Publicly available datasets from Airly sensors were analyzed in this study and can be found here: (https://map.airly.org/, accessed on 17 February 2025). API documentation from Airly is available here: (https://developer.airly.org/en/docs, accessed on 17 February 2025). Publicly available datasets from the Chief Inspectorate For Environmental Protection database were analyzed in this study. This data can be found here: (http://powietrze.gios.gov.pl/pjp/home, accessed on 17 February 2025). API documentation is available here: (http://powietrze.gios.gov.pl, accessed on 17 February 2025). Additional data that support the findings of this study are available from the corresponding author, Michał Lupa, upon reasonable request.

**Acknowledgments:** We would like to express our gratitude to the Małopolska Provincial Office in Krakow for their ongoing collaboration and for providing historical data for our analyses. Special thanks are extended to Piotr Kubik for his invaluable contribution to the analyses and his significant input on the subject matter.

**Conflicts of Interest:** The authors declare no conflicts of interest.

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
