# Peer review of "Emergency Medical Interventions in Areas with High Air Pollution: A Case Study from Małopolska Voivodeship, Poland"

_atmosphere, doi:10.3390/atmos16080983_

Round 1
Reviewer 1 Report
Comments and Suggestions for Authors
This study links air pollution to EMS calls in Malopolska, Poland (2020–2023). PM10 data from 192 sensors shows more EMS calls, especially respiratory, during high pollution. Most municipalities exceeded PM10 limits.
1.The methodology section provides a detailed description of data sources and spatial aggregation techniques. However, the explanation of the Bayesian hierarchical model is insufficient. Please include a clearer description of the model structure, including the priors used and how they were justified, to enhance reproducibility and transparency.
2.:The study acknowledges data availability issues, particularly with Airly sensors in 2023, but does not sufficiently discuss how these gaps may have impacted the results. A more robust discussion on potential biases, especially regarding the incomplete temporal coverage and sensor noise from LookO2.
3. The analysis of COVID-19's influence on EMS call volumes is a critical component, but the manuscript could benefit from a deeper exploration of how lockdown-related air quality improvements were accounted for in the statistical models. Consider including a sensitivity analysis to quantify the effect of excluding pandemic periods on the observed correlations.
4.In the introduction, it is evident that the authors do not have sufficient information on existing studies. It is suggested that the authors should explore more studies by others. For instance, refer to ‘VAR - tree model based spatio - temporal characterization and prediction of O3 concentration in China’ and ‘Multi - objective optimal dispatch strategy for power systems with Spatio - temporal distribution of air pollutants’.
5.The use of the Kolmogorov-Smirnov test is appropriate, but the manuscript would be improved by including additional statistical methods to validate the findings, such as regression models to quantify the strength of the relationship between PM10 exceedances and EMS calls.
Author Response
We sincerely appreciate the Reviewers' insightful comments and suggestions, which have been valuable in further enhancing the quality of our manuscript.
To better observe communicating our response, we divided our responses into three categories: Agree/Clarification/Disagree.
Responses to the Reviewer#1
Suggestion, Question, or Comment from the Reviewer#2 |
Author’s Response |
Change in the Manuscript |
1.The methodology section provides a detailed description of data sources and spatial aggregation techniques. However, the explanation of the Bayesian hierarchical model is insufficient. Please include a clearer description of the model structure, including the priors used and how they were justified, to enhance reproducibility and transparency. |
Clarification |
We would like to clarify that no Bayesian hierarchical model was applied in this study. Our analysis was based on classical statistical methods, including non-parametric Kolmogorov-Smirnov tests to compare distributions of EMS call volumes under different air quality conditions, Pearson correlation analysis to evaluate relationships between air pollution and EMS call frequency and comparative analysis of municipalities categorized by long-term air quality. |
2.The study acknowledges data availability issues, particularly with Airly sensors in 2023, but does not sufficiently discuss how these gaps may have impacted the results. A more robust discussion on potential biases, especially regarding the incomplete temporal coverage and sensor noise from LookO2. |
Agree |
We have expanded the Discussion section to address these limitations in more detail. Specifically, we now discuss the incomplete temporal coverage of Airly sensors in 2023 and the higher noise levels observed in LookO2 data, while clarifying that their impact on the overall multi-year results is minimal. |
3. The analysis of COVID-19's influence on EMS call volumes is a critical component, but the manuscript could benefit from a deeper exploration of how lockdown-related air quality improvements were accounted for in the statistical models. Consider including a sensitivity analysis to quantify the effect of excluding pandemic periods on the observed correlations. |
Clarification |
According to an analysis published by Airly - the largest provider of air quality sensors in Europe data do not indicate a substantial improvement in air quality across the continent during the pandemic. Despite widespread perceptions and some early publications suggesting reductions in pollutant levels, Airly explicitly states as it is in their report available at: https://airly.org/pl/czy-koronawirus-wplynal-na-zanieczyszczenia-powietrza-w-europie-nasza-analiza/ In light of this, and consistent with our own dataset, we reaffirm that the observed relationships between PM10 exceedances and EMS call volumes are unlikely to be driven by lockdown‑related improvements in air quality. |
4.In the introduction, it is evident that the authors do not have sufficient information on existing studies. It is suggested that the authors should explore more studies by others. For instance, refer to ‘VAR - tree model based spatio - temporal characterization and prediction of O3 concentration in China’ and ‘Multi - objective optimal dispatch strategy for power systems with Spatio - temporal distribution of air pollutants’. |
Agree |
Both recommended references have been added to the Introduction to strengthen the context of existing research. |
5.The use of the Kolmogorov-Smirnov test is appropriate, but the manuscript would be improved by including additional statistical methods to validate the findings, such as regression models to quantify the strength of the relationship between PM10 exceedances and EMS calls. |
Clarification |
Thanks for the suggestion and appreciation. In our other research - that we added references in the manuscript - we used not only Two-Stage Least Squares (2SLS) regression with bootstrapping to examine the influence of energy mix on air pollution but also standardized GWR, local Moran’s I spatial autocorrelation and Getis-Ord Gi* statistics to evaluate the influence of meteorological factors and terrain on air pollution concentration and migration and also we used However, for our current study, regression tests would not align with the main aims of the research. In this research we wanted to focus more on the calls and pollution with spatial factors. The relation, as we appreciate, between these two measurands is not straightforward. The aim of this limited study is to understand if at a generative model level these two variables are related to each other. This has been shown using K-S tests. The exact nature of this relationship and how best to capture it would be the matter of investigation of future study which would also need pertinent measurement campaigns. We will be happy to include your suggestion in future research but for now to avoid replication of previous ones and to not make simple correlation between 2 variables that in our opinion may be misleading we would like to keep the methods as presented. We hope it is satisfactory for you. |
Reviewer 2 Report
Comments and Suggestions for Authors
See attachments.

Author Response
We sincerely appreciate the Reviewers' insightful comments and suggestions, which have been valuable in further enhancing the quality of our manuscript.
To better observe communicating our response, we divided our responses into three categories: Agree/Clarification/Disagree.
Responses to the Reviewer#2
Suggestion, Question, or Comment from the Reviewer#2 |
Author’s Response |
Change in the Manuscript |
Point 1. Include at least one or two numerical results in the abstract to strengthen its quantitative significance |
Agree |
The abstract has been updated to include specific numerical results, indicating the total number of EMS calls analyzed (305,142) and the average number of respiratory-related EMS calls per 1,000 residents in months with and without PM10 exceedances (1.50 vs. 1.19). These additions strengthen the quantitative impact of the abstract while maintaining its clarity. |
Point 2. Introduction section expand context and gaps. The pathophysiological link between PM10 exposure and cardiovascular/respiratory emergencies is oversimplified. Add 1–2 sentences on how regional policies (e.g., coal bans) or socioeconomic factors exacerbate pollution |
Agree |
The introduction section has been significantly expanded to include a more detailed discussion of the pathophysiological links between PM₁₀ exposure and cardiovascular and respiratory emergencies. |
Point 3. Chemical notation formatting. Inconsistent formatting appears throughout: PM10 (correct), O2 (should be O₂), and specific pollutants (e.g., NO₂, SO₂). Define all abbreviations at first use (e.g., "particulate matter ≤10 μm, PM₁₀"). |
Agree |
We have thoroughly reviewed the manuscript to ensure consistent and correct formatting of chemical notations and abbreviations throughout the text. |
Point 4. Missing subsection: No dedicated study area description. Insert "Section2.1 Study Area Overview" before air pollution data sources. Include: Geographic/topographic traits of Malopolska (e.g., basins causing inversion), Demographics, key pollution sources (e.g., residential coal use), and administrative divisions. |
Agree |
A new subsection 2.1 Study Area Overview has been added before the description of air pollution data sources. |
Point 5. Add a location map (Poland inset + Malopolska detail) highlighting high/low-pollution municipalities (align with Table 1/2). |
Clarification |
A location map has been added to Section 2.1 Study Area Overview, showing the Małopolska Voivodeship within Poland and the counties included in the analysis. Regarding the request to additionally highlight high- and low-pollution municipalities on this map, we kindly note that subsequent figures in the 2.4 section (Figures 3 and 4) already present detailed spatial distributions of PM10 exceedances and classification of municipalities according to pollution levels. To avoid redundancy and maintain the concise structure of the manuscript, we propose not to add another figure duplicating this information. Moreover, we consider that Section 2.1, which is dedicated to describing the study area, should remain neutral and should not pre-empt the results by indicating which municipalities are most or least polluted. |
Point 6. Section 2: Justify statistical methods. Statistical tests (Kolmogorov-Smirnov, Pearson) are named but not rationalized. Add a brief justification (1–2 sentences per method) in 2.5 Statistical Analysis: |
Agree |
In Section 2.5, we have now added a brief justification for the selected statistical methods. |
Point 7. Figure 4: Add error bars (±SD) to each bar. Specify if data represents means ± SD in the caption. |
Clarification |
We thank the Reviewer for this suggestion. Figure 4 presents annual totals of EMS calls, and each bar represents a single yearly sum. Since these totals do not include repeated measurements, standard deviations cannot be meaningfully calculated or visualized. We believe that the current representation accurately reflects the nature of the data. |
Point 8. In Result Section, (1) Spatial results are vague (e.g., "higher EMS calls in polluted municipalities"). Quantify differences (e.g., *"Municipalities with poor air quality had 12.1% more respiratory calls (1.28 vs. 1.12 calls/1000 people, p<0.001)"*). (3) Table 3.1 (p. 10) uses inconsistent numbering (Table 3 vs. Table 3.1). Rename as Table 3 and relabel subtables (e.g., Table 3a, 3b). |
Agree |
We have revised the Results section to provide quantitative details for the spatial analysis. Specifically, we report that municipalities with poor air quality experienced 12.1% more respiratory EMS calls on average (1.28 vs. 1.12 per 1,000 residents, p < 0.001), with a smaller difference observed for cardiovascular calls (0.58 vs. 0.56 per 1,000 residents, p = 0.0008). Section 3.4.1 now clarifies the definition of “COVID-19 peak periods,” which were identified based on national daily case maxima using a quadratic trend model, following Li & Linton (2020). Finally, Table 3 has been renumbered and split into sub-tables (3a, 3b) for clarity. |
Point 9. In “Discussion” Section, no comparison of effect sizes with prior studies. Contrast key findings (e.g., *"Our KS statistic for respiratory calls (0.249) exceeded McLeod et al.’s reported correlation (r=0.18), suggesting finer spatial resolution strengthens pollution-health linkages"*). |
Agree |
We have revised the Discussion section to include a comparison of our observed effect sizes with prior studies as suggested |
Point 10. Add a "5. Conclusion" section summarizing |
Agree |
We have added a new Section 5. Conclusions, which provides a concise summary of the study’s key findings and their practical implications for public health and EMS planning. |
Reviewer 3 Report
Comments and Suggestions for Authors
- Title of section 3 should be named as analyses and results, then it would be more suitable to the context of this section. Same as section 4, which would be better titles as conclusion and discussion.
- Section 2 focus on air pollution and its spatial and temporal features and trend. Then the question is which section are focused on the EMS besides the detail discussion in section 3.
- By the way, the methodology taken as threshold-based approach in the detail analysis the relationship between air pollution and EMS, should be discussed its applicability.
Author Response
We sincerely appreciate the Reviewers' insightful comments and suggestions, which have been valuable in further enhancing the quality of our manuscript.
To better observe communicating our response, we divided our responses into three categories: Agree/Clarification/Disagree.
Responses to the Reviewer#3
Suggestion, Question, or Comment from the Reviewer#1 |
Author’s Response |
Change in the Manuscript |
Title of section 3 should be named as analyses and results, then it would be more suitable to the context of this section. Same as section 4, which would be better titles as conclusion and discussion. |
Agree |
Thank you for your suggestion. We have decided to follow the official MDPI template, which uses the standard structure “3. Results” and “4. Discussion”. To address your concern regarding the conclusions, we have now included a separate Section 5. Conclusions, which provides a concise summary of the study’s key findings and practical implications. This structure ensures clarity while remaining consistent with the journal’s formatting guidelines. |
Section 2 focus on air pollution and its spatial and temporal features and trend. Then the question is which section are focused on the EMS besides the detail discussion in section 3. |
Agree |
We have clarified the description of EMS data in a new Section 2.5 EMS Data Processing. |
By the way, the methodology taken as threshold-based approach in the detail analysis the relationship between air pollution and EMS, should be discussed its applicability. |
Agree |
We have added a justification for the threshold-based approach at the end of Section 2.3 Pollution Assessment Methods. |
Reviewer 4 Report
Comments and Suggestions for Authors
The authors present a study associating air pollution (PM10) with emergency medical interventions. They acknowledge the limitation provided by COVID-19 infection waves and investigate seasonality. The document is well written but there are some presentation concerns listed below. My main suggestion, would be for the authors to obtain and analyze COVID-19 infections or mortality rates during the wave periods highlighted. This could be used as a normalizer as opposed to assuming homogeneity across the time period. This may help clarify some of the synergistic pollution, seasonality, and COVID-19 impacts on EMS frequency.
Line 13: Generally "10" in "PM10" is subscripted.
Line 33: Fine particulate matter only refers to PM2.5. PM10 is considered "coarse" particulate matter. Both should be subscripted as well (and elsewhere throughout the document).
Lines 84-7: The sensor description should be accompanied with citations for the reader to learn more about the technology used. From this paragraph, it seems that the sensors are not all of the same quality which can skew results. The description of the validation needs much more development than is presented here as the sensors don't seem to be co-located so there must be a clear description of how to ensure their quality is uniform.
Lines 111-2: It seems that the authors only studied daily averages. Would the results be different if they also studied hourly peaks? There is a substantial body of literature (particularly for short term events) where hourly peaks have a greater health impact than daily averages.
Line 142: Please clarify the spatial (and/or population) differences (or ranges) between districts, municipalities, and counties.
Figure 2: The scale bar would benefit from using only whole numbers. Also (while obvious) it would be helpful to explicitly list "latitude" and "longitude" along with degree symbols (this applies to all maps).
Line 160: Please explain service-oriented and historical-monumental.
Line 180: It seems that if (for example) one municipality has 18 exceedance days in 180 days of valid data, it could reasonably be assumed that it may exceed the 35-day limit.
Table 2: The table looks to have smaller font than Table 1. Also "Suma" may be "Sum".
Figure 3: Is there a color scale for the right plot? Also, subplots should be labeled "(a)", "(b)", etc.
Figure 4: Consider using a comma delimiter for the y-axis labels.
Line 240: Why was 2023 chosen as the population year?
Line 268: This should just be "Table 3".
Lines 325-9: Consider being consistent using lists. Here they are letters ("a", "b"), in 3.2 they are numbers, and in 3.1 they are bullet points.
Figure 8: The legend on the top left should be on top of the axis lines.
Line 488: Type S is not a globally recognized ambulance type. Consider providing an alternative name.
Comments on the Quality of English Language
The English language is relatively good - I listed a few issues in the comments but a full proofread should address them.
Author Response
We sincerely appreciate the Reviewers' insightful comments and suggestions, which have been valuable in further enhancing the quality of our manuscript.
To better observe communicating our response, we divided our responses into three categories: Agree/Clarification/Disagree.
Responses to the Reviewer#4
Suggestion, Question, or Comment from the Reviewer#1 |
Author’s Response |
Change in the Manuscript |
My main suggestion, would be for the authors to obtain and analyze COVID-19 infections or mortality rates during the wave periods highlighted. This could be used as a normalizer as opposed to assuming homogeneity across the time period. This may help clarify some of the synergistic pollution, seasonality, and COVID-19 impacts on EMS frequency. |
Clarification |
We appreciate the reviewer’s suggestion to normalize EMS call volumes using COVID‑19 infections or mortality rates during the identified pandemic waves. Unfortunately, publicly available COVID‑19 datasets in Poland do not provide sufficiently granular data at the county level, which is the primary spatial unit used in our analysis. National data available on request but no official dataset exists that consistently reports confirmed cases or deaths at the level for the entire study period. We also examined archived governmental reports and regional summaries; however, they either aggregate cases at the voivodeship level or are published in formats unsuitable for continuous daily or monthly time series analysis. Consequently, integrating such data as a normalizer for EMS calls was not feasible. In light of these limitations, our study interprets the interaction between air pollution and EMS demand without direct COVID‑19 incidence normalization, while acknowledging this as a limitation for future research. |
Line 13: Generally "10" in "PM10" is subscripted. |
Agree |
Corrected. |
Line 33: Fine particulate matter only refers to PM2.5. PM10 is considered "coarse" particulate matter. Both should be subscripted as well (and elsewhere throughout the document). |
Agree |
The sentence was revised to more accurately distinguish between fine particulate matter. |
Lines 84-7: The sensor description should be accompanied with citations for the reader to learn more about the technology used. From this paragraph, it seems that the sensors are not all of the same quality which can skew results. The description of the validation needs much more development than is presented here as the sensors don't seem to be co-located so there must be a clear description of how to ensure their quality is uniform. |
Agree |
We added description in line with WMO recommendation and references to other studies that are evaluating LCS sensors used in this study in the context of reference measurements |
Lines 111-2: It seems that the authors only studied daily averages. Would the results be different if they also studied hourly peaks? There is a substantial body of literature (particularly for short term events) where hourly peaks have a greater health impact than daily averages. |
Clarification |
Thank you for your comment. You are absolutely right that there is a substantial body of literature suggesting that short-term peaks in air pollution (e.g., hourly concentrations) may have a stronger association with acute health outcomes than daily averages. In our study, we used daily averages of PM indicators and ambulance dispatches, as the health data were only available at a daily resolution. Therefore, analyzing hourly peaks was not feasible within the scope of this dataset. However, we would like to emphasize that even using daily averages, it is possible to distinguish between days with only background PM levels and those with noticeable pollution peaks. This distinction is still reflected in the daily mean values, which allows us to capture relevant variation and detect associations with health outcomes. At this stage, we do not plan to modify our methodological approach, but we agree that exploring finer temporal scales would be a valuable direction for future studies, particularly where high-resolution health data are available. |
Line 142: Please clarify the spatial (and/or population) differences (or ranges) between districts, municipalities, and counties. |
Agree |
We have clarified the spatial and population differences between districts, municipalities, and counties in Section 2.1 Study Area Overview. |
Figure 2: The scale bar would benefit from using only whole numbers. Also (while obvious) it would be helpful to explicitly list "latitude" and "longitude" along with degree symbols (this applies to all maps). |
Agree |
The map figures have been updated according to the reviewer’s suggestions. Scale bars now display only whole numbers, and all maps explicitly include “Latitude” and “Longitude” labels with degree symbols. |
Line 160: Please explain service-oriented and historical-monumental. |
Agree |
Thank you for the suggestion regarding the clarity of land-use terminology. In the revised manuscript, the term “service-oriented” has been replaced with “commercial”, and “historical-monumental” has been simplified to “historical” to avoid potential ambiguity and improve readability. |
Line 180: It seems that if (for example) one municipality has 18 exceedance days in 180 days of valid data, it could reasonably be assumed that it may exceed the 35-day limit. |
Clarification |
Thank you for this comment. We are aware of this limitation and have explicitly noted it in the limitations. In cases where measurements were available for a reduced period (e.g., 180 valid days), there is a possibility that the total annual exceedances could surpass the 35-day regulatory threshold if the remaining months were also monitored. So, there were two possible approaches:
or
We chose the latter because missing months often correspond to spring–summer periods, which typically exhibit fewer exceedances of PM₁₀ in Małopolska. Proportional extrapolation would risk overestimating annual exceedances in such cases. |
Table 2: The table looks to have smaller font than Table 1. Also "Suma" may be "Sum". |
Agree |
Corrected. |
Figure 3: Is there a color scale for the right plot? Also, subplots should be labeled "(a)", "(b)", etc. |
Agree |
We have revised the figures according to the reviewer’s suggestions. |
Figure 4: Consider using a comma delimiter for the y-axis labels. |
Agree |
We have revised the figure according to the reviewer’s suggestions. |
Line 240: Why was 2023 chosen as the population year? |
Clarification |
We have clarified in the manuscript that 2023 was chosen as the population reference year because demographic changes in Małopolska between 2020 and 2023 were minimal (<1%). According to Statistics Poland reports the population decreased from 3.41 to 3.39 million (≈0.5%), which does not materially influence EMS call rates per 1,000 residents. Urząd Statystyczny w Krakowie. Sytuacja demograficzna województwa małopolskiego w 2023 r.; 2024. Available online: https://krakow.stat.gov.pl (accessed on 6 August 2025). |
Line 268: This should just be "Table 3". |
Agree |
Corrected. |
Lines 325-9: Consider being consistent using lists. Here they are letters ("a", "b"), in 3.2 they are numbers, and in 3.1 they are bullet points. |
Agree |
All lists were unified and have bullet points now. |
Figure 8: The legend on the top left should be on top of the axis lines. |
Agree |
Corrected. |
Line 488: Type S is not a globally recognized ambulance type. Consider providing an alternative name. |
Agree |
Type S has been changed to ALS |
Round 2
Reviewer 1 Report
Comments and Suggestions for Authors
It is recommended to accept directly.
Author Response
We sincerely thank the reviewer for the positive evaluation of our work and the recommendation for direct acceptance.
Reviewer 2 Report
Comments and Suggestions for Authors
-
(Lines 93–102) The introduction outlines the study’s aims but does not clearly articulate how it advances beyond prior literature, particularly McLeod et al. [23] and Wanka et al. [24]. Please explicitly state the methodological or contextual innovation (e.g., spatial granularity, integration of multi-source sensors, pandemic adjustment) and how these improve upon existing approaches.
-
(Lines 17, 46–47, 63–65) The formatting of PM₁₀ is inconsistent (e.g., “PM1ń0” in line 17 appears to be a typographical error). Please ensure all pollutant symbols use standard subscript formatting for clarity.
-
(Lines 168–177) While outlier removal is mentioned, the manuscript lacks detail on calibration between sensor types. Since LookO₂ sensors exhibited higher noise, please describe any inter-calibration procedures or statistical harmonization applied to mitigate systematic bias.
-
(Lines 182–184) The concentration of sensors in Kraków versus rural gaps (e.g., Limanowa) could bias spatial comparisons. Please quantify potential bias or test robustness by excluding oversampled urban areas to assess whether results hold.
-
(Lines 229–236) The use of “any sensor above 50 µg/m³ defines exceedance for the municipality” may overestimate exposure if only one sensor records an extreme value. Consider providing sensitivity analyses using mean values or requiring multiple sensors to exceed the threshold.
-
(Lines 195–199) The “respiratory” category includes chest pain, which is often cardiovascular in origin. Please justify this classification choice or provide results with chest pain reassigned to cardiovascular to test robustness.
-
(Lines 367–395) While the KS test compares distributions, the text sometimes implies magnitude comparison rather than distributional differences. Please clarify this distinction and ensure interpretations align with the test’s statistical meaning.
-
(Lines 432–484) The pandemic’s overlap with winter pollution seasons is acknowledged, but the adjustment by removing “peak” months may also remove a disproportionate number of high-pollution observations. Consider alternative adjustments (e.g., including COVID-19 incidence as a covariate in regression models) to retain seasonality information.
-
(Lines 488–495) The quarterly aggregation in Figure 10 may be too coarse to detect seasonal cycles. Monthly or even weekly seasonal decomposition (e.g., STL) could more clearly separate pollution-driven effects from pandemic noise.
-
(Lines 596–597) The absence of meteorological variables (wind speed, temperature inversion data) is an important limitation. Consider discussing how their inclusion could alter pollution–EMS relationships, especially in a topographically complex area like Małopolska.
-
(Lines 622–626) The suggestion to allocate ALS units to high-pollution municipalities is interesting but currently speculative. Please support this with a quantitative estimate of potential resource needs or health benefits, or frame it more cautiously.
-
Several figures use color scales without clear legends or numerical ranges, making interpretation difficult for color-blind readers. Please ensure all maps/plots have labeled color bars and, where possible, include numeric scales or categories.
It is recommended that the author should check the entire manuscript for English and grammar mistakes.
Author Response
We sincerely appreciate the Reviewers' insightful comments and suggestions, which have been valuable in further enhancing the quality of our manuscript.
To better observe communicating our response, we divided our responses into three categories: Agree/Clarification/Disagree.
Suggestion, Question, or Comment from the Reviewer#2 |
Author’s Response |
Change in the Manuscript |
(Lines 93–102) The introduction outlines the study’s aims but does not clearly articulate how it advances beyond prior literature, particularly McLeod et al. [23] and Wanka et al. [24]. Please explicitly state the methodological or contextual innovation (e.g., spatial granularity, integration of multi-source sensors, pandemic adjustment) and how these improve upon existing approaches. |
Agree |
We have revised the Introduction to explicitly state our methodological and contextual contributions. |
(Lines 17, 46–47, 63–65) The formatting of PM₁₀ is inconsistent (e.g., “PM1ń0” in line 17 appears to be a typographical error). Please ensure all pollutant symbols use standard subscript formatting for clarity. |
Agree |
All pollutant symbols have been unified. |
(Lines 168–177) While outlier removal is mentioned, the manuscript lacks detail on calibration between sensor types. Since LookO₂ sensors exhibited higher noise, please describe any inter-calibration procedures or statistical harmonization applied to mitigate systematic bias. |
Agree |
Thank you for pointing this out. We have clarified how cross-network differences were handled. We did not perform a formal inter-calibration across GIOŚ, Airly, and LookO₂. Instead, we minimized scale differences by (i) computing daily means after sensor-level outlier filtering, and (ii) defining exposure as municipality-level PM₁₀ exceedance using an any-sensor >50 μg/m³ |
(Lines 182–184) The concentration of sensors in Kraków versus rural gaps (e.g., Limanowa) could bias spatial comparisons. Please quantify potential bias or test robustness by excluding oversampled urban areas to assess whether results hold. |
Clarification |
We acknowledge the concern about unequal sensor density. To limit potential bias from Kraków’s dense network, we analysed Kraków at district level (sub-county units) rather than as a single county, so its sensors are distributed across several spatial units. All other areas were analysed at the municipality level. This design prevents the Kraków network from dominating spatial comparisons while retaining policy-relevant administrative units. |
(Lines 229–236) The use of “any sensor above 50 µg/m³ defines exceedance for the municipality” may overestimate exposure if only one sensor records an extreme value. Consider providing sensitivity analyses using mean values or requiring multiple sensors to exceed the threshold. |
Clarification |
Thank you for the suggestion. In preliminary analyses, we also computed municipality-level monthly means (averaging across all available sensors in a municipality) and obtained qualitatively consistent spatial patterns and associations. We ultimately adopted a threshold-based definition (PM₁₀ > 50 µg/m³) because it better captures episodic winter pollution and is less sensitive to cross-network scale differences than continuous concentrations. |
(Lines 195–199) The “respiratory” category includes chest pain, which is often cardiovascular in origin. Please justify this classification choice or provide results with chest pain reassigned to cardiovascular to test robustness. |
Agree |
Thank you for catching this. This was a translation error: the intended item was pleuritic chest pain (i.e., chest pain on inspiration/with breathing), which is a respiratory symptom, not generic chest pain. We have corrected the English wording and clarified the category definitions. In our coding, unspecified chest pain / suspected ACS is classified under cardiovascular, whereas pleuritic chest pain, dyspnea, wheezing, and cough remain under respiratory. |
(Lines 367–395) While the KS test compares distributions, the text sometimes implies magnitude comparison rather than distributional differences. Please clarify this distinction and ensure interpretations align with the test’s statistical meaning. |
Agree |
We clarified that the KS test evaluates distributional (ECDF) differences rather than mean magnitude. We added an explicit sentence in Methods (Sec. 2.7) describing how KS results are interpreted, and we revised the wording in Results to refer to distributional shifts (e.g., right/left shift, stochastically larger/smaller) instead of higher/lower numbers. |
(Lines 432–484) The pandemic’s overlap with winter pollution seasons is acknowledged, but the adjustment by removing “peak” months may also remove a disproportionate number of high-pollution observations. Consider alternative adjustments (e.g., including COVID-19 incidence as a covariate in regression models) to retain seasonality information. |
Clarification |
Thank you for the comment. Our primary analyses use the full 2020–2023 sample (see Table 3); the exclusion of peak pandemic windows was applied only as a robustness check to reduce demand-side shocks (Tables 5–6). As seasonality is a substantial topic on its own, we plan a separate follow-up paper to model seasonal structure in detail (e.g., monthly/weekly decomposition), which is beyond the scope of the current non-parametric design. |
(Lines 488–495) The quarterly aggregation in Figure 10 may be too coarse to detect seasonal cycles. Monthly or even weekly seasonal decomposition (e.g., STL) could more clearly separate pollution-driven effects from pandemic noise. |
Agree |
Figure and table were changed to show monthly aggregation. |
(Lines 596–597) The absence of meteorological variables (wind speed, temperature inversion data) is an important limitation. Consider discussing how their inclusion could alter pollution–EMS relationships, especially in a topographically complex area like Małopolska. |
Agree |
We agree with the reviewer that the absence of meteorological and topographic variables is an important limitation. We’ve divided the discussion section and added a subsection where all of the limitations have been discussed. |
(Lines 622–626) The suggestion to allocate ALS units to high-pollution municipalities is interesting but currently speculative. Please support this with a quantitative estimate of potential resource needs or health benefits, or frame it more cautiously. |
Agree |
Thank you for your comment. We have revised the sentence to frame the suggestion more cautiously, emphasizing that it is a potential implication rather than a definitive recommendation, and noting the need for further quantitative assessment before implementation. |
Several figures use color scales without clear legends or numerical ranges, making interpretation difficult for color-blind readers. Please ensure all maps/plots have labeled color bars and, where possible, include numeric scales or categories. |
Agree |
All plots and maps have been labelled according to the reviewers’ suggestion. |
Comments on the Quality of English Language — It is recommended that the author should check the entire manuscript for English and grammar mistakes. |
Agree |
We have revised manuscript in order to improve the language quality. |
Reviewer 3 Report
Comments and Suggestions for Authors
The suggestions have been accepted and presented properly which are presented for last version.
This version of this article can be accepted in present form.
Author Response

(The authors gave the same response as above.)

Reviewer 4 Report
Comments and Suggestions for Authors The authors have substantially improved the document. There are a few outstanding issues: Original comment #1: "Lines 111-2: It seems that the authors only studied daily averages. Would the results be different if they also studied hourly peaks? There is a substantial body of literature (particularly for short term events) where hourly peaks have a greater health impact than daily averages." This was partially addressed by the authors - while the methodology does not need to be changed, they should consider listing (most likely in the Discussion section) as future work the use of hourly (or similar) peaks instead of daily averages. It may be a good idea to subdivide the Discussion into Findings, Limitations, and Future Work (or similar names) as it is relatively long and commingles all these topics.Original comment #2:
"Figure 2: The scale bar would benefit from using only whole numbers. Also (while obvious) it would be helpful to explicitly list "latitude" and "longitude" along with degree symbols (this applies to all maps)."
In Figures 2 and 3, the x and y-axes have repeated numbers and they should be fixed. I think the authors misunderstood my comment. I meant that the scale bar (e.g., 0-20 in Figure 3) should be whole numbers, for example 0, 2, 4...20 not that the Latitude and Longitude numbers should be whole numbers. However, I suggest they, for example, start at 49.25, then 49.50, etc. (for the Latitude, and similarly for the Longitude). This applies to all figures. Figure 1 does not include latitude and longitude. Original comment #3: "Table 2: The table looks to have smaller font than Table 1. Also "Suma" may be "Sum"." Table still has a column labeled "Suma". New Comments: - Why is Figure 4.b. smaller than 4.a.? - It would be a good idea to ensure that all maps have a similar latitude/longitude scale. It seems that Figure 6 covers a different extent. - Figure 7 legend (and elsewhere) consider subscripting "10" in "PM10".
Author Response
We sincerely appreciate the Reviewers' insightful comments and suggestions, which have been valuable in further enhancing the quality of our manuscript.
To better observe communicating our response, we divided our responses into three categories: Agree/Clarification/Disagree.
Suggestion, Question, or Comment from the Reviewer#4 |
Author’s Response |
Change in the Manuscript |
Original comment #1: "Lines 111-2: It seems that the authors only studied daily averages. Would the results be different if they also studied hourly peaks? There is a substantial body of literature (particularly for short term events) where hourly peaks have a greater health impact than daily averages." This was partially addressed by the authors – while the methodology does not need to be changed, they should consider listing (most likely in the Discussion section) as future work the use of hourly (or similar) peaks instead of daily averages. It may be a good idea to subdivide the Discussion into Findings, Limitations, and Future Work (or similar names) as it is relatively long and commingles all these topics. |
Agree |
Added a note in the “Future Work” subsection about using hourly peaks in future analyses. Restructured Discussion into Findings, Limitations, and Future Work. |
Original comment #2: "Figure 2: The scale bar would benefit from using only whole numbers. Also (while obvious) it would be helpful to explicitly list 'latitude' and 'longitude' along with degree symbols (this applies to all maps)." In Figures 2 and 3, the x and y-axes have repeated numbers and they should be fixed. I think the authors misunderstood my comment. I meant that the scale bar (e.g., 0–20 in Figure 3) should be whole numbers, for example 0, 2, 4…20 not that the Latitude and Longitude numbers should be whole numbers. However, I suggest they, for example, start at 49.25, then 49.50, etc. (for the Latitude, and similarly for the Longitude). This applies to all figures. Figure 1 does not include latitude and longitude. |
Agree |
Revised all figures to use whole-number scale bars, proper degree notation, adjusted increments, and corrected duplicated axis numbers. Added latitude/longitude to Figure 1. |
Original comment #3: "Table 2: The table looks to have smaller font than Table 1. Also 'Suma' may be 'Sum'." Table still has a column labeled "Suma". |
Agree |
Changed “Suma” to “Sum” and matched font type and size to Table 1. |
New comment: Why is Figure 4.b smaller than 4.a.? |
Agree |
Adjusted Figures 4.a and 4.b to equal dimensions. |
New comment: It would be a good idea to ensure that all maps have a similar latitude/longitude scale. It seems that Figure 6 covers a different extent. |
Agree |
Standardized latitude/longitude scales and map extents across figures except Fig. 1. which shows the general location of the analyzed region. |
New comment: Figure 7 legend (and elsewhere) consider subscripting "10" in "PM10". |
Agree |
Updated pollutant notation to use proper subscript formatting (PM₁₀) in all relevant locations. |